# Strategies in the Design and Development of Non-Nucleoside Reverse Transcriptase Inhibitors (NNRTIs)

**DOI:** 10.3390/v15101992

**Published:** 2023-09-25

**Authors:** Murugesan Vanangamudi, Senthilkumar Palaniappan, Muthu Kumaradoss Kathiravan, Vigneshwaran Namasivayam

**Affiliations:** 1Department of Pharmaceutical Chemistry, Amity Institute of Pharmacy, Amity University Madhya Pradesh, Gwalior 474005, Madhya Pradesh, India; vmurugesan@gwa.amity.edu; 2Faculty of Pharmacy, Karpagam Academy of Higher Education, Coimbatore 641021, Tamilnadu, India; drsenthilkumar.p@kahedu.edu.in; 3Center for Active Pharmaceutical Ingredients, Karpagam Academy of Higher Education, Coimbatore 641021, Tamilnadu, India; 4Dr. APJ Abdul Kalam Research Lab, SRM College of Pharmacy, SRMIST, Kattankulathur 603203, Tamilnadu, India; kathirak@srmist.edu.in; 5Department of Pharmaceutical Chemistry, SRM College of Pharmacy, SRMIST, Kattankulathur 603203, Tamilnadu, India; 6Pharmaceutical Chemistry, Pharmaceutical Institute, University of Bonn, 53121 Bonn, Germany; 7LIED, University of Lübeck and University Medical Center Schleswig-Holstein, Ratzeburger Allee 160, 23538 Lübeck, Germany

**Keywords:** HIV/AIDS, NNRTIs, structure- and fragment-based drug design, bioisosteric replacement, prodrug, scaffold hopping, molecular hybridization, cART, long-acting injectable, pharmaceutical strategies, reverse transcriptase

## Abstract

AIDS (acquired immunodeficiency syndrome) is a potentially life-threatening infectious disease caused by human immunodeficiency virus (HIV). To date, thousands of people have lost their lives annually due to HIV infection, and it continues to be a big public health issue globally. Since the discovery of the first drug, Zidovudine (AZT), a nucleoside reverse transcriptase inhibitor (NRTI), to date, 30 drugs have been approved by the FDA, primarily targeting reverse transcriptase, integrase, and/or protease enzymes. The majority of these drugs target the catalytic and allosteric sites of the HIV enzyme reverse transcriptase. Compared to the NRTI family of drugs, the diverse chemical class of non-nucleoside reverse transcriptase inhibitors (NNRTIs) has special anti-HIV activity with high specificity and low toxicity. However, current clinical usage of NRTI and NNRTI drugs has limited therapeutic value due to their adverse drug reactions and the emergence of multidrug-resistant (MDR) strains. To overcome drug resistance and efficacy issues, combination therapy is widely prescribed for HIV patients. Combination antiretroviral therapy (cART) includes more than one antiretroviral agent targeting two or more enzymes in the life cycle of the virus. Medicinal chemistry researchers apply different optimization strategies including structure- and fragment-based drug design, prodrug approach, scaffold hopping, molecular/fragment hybridization, bioisosterism, high-throughput screening, covalent-binding, targeting highly hydrophobic channel, targeting dual site, and multi-target-directed ligand to identify and develop novel NNRTIs with high antiviral activity against wild-type (WT) and mutant strains. The formulation experts design various delivery systems with single or combination therapies and long-acting regimens of NNRTIs to improve pharmacokinetic profiles and provide sustained therapeutic effects.

## 1. Introduction

Acquired immunodeficiency syndrome (AIDS) is caused by the Human Immunodeficiency Virus (HIV) which suppresses the human CD4+ T-cell immune system supporting opportunistic infections. Currently, HIV is one of the most infectious diseases in the world, as there is neither a successful vaccine nor a drug that can completely cure or eradicate the disease [1]. Reverse transcriptase (RT), integrase, and protease are the major enzymes that play a key role in the development of the matured virus in an HIV-infected patient [2]. RT inhibitors are one of the main important components in combination antiretroviral therapy (cART) to decrease AIDS-related mortality [3]. Drugs binding to the RT enzyme are classified into two categories, namely nucleoside and non-nucleoside inhibitors, based on the binding sites of catalytic and hydrophobic allosteric sites of the RT enzyme. The combination regimen comprises mainly a non-nucleoside reverse transcriptase inhibitor (NNRTI) along with a nucleoside reverse transcriptase inhibitor (NRTI), and a protease inhibitor (PI) or an integrase inhibitor (INSTI). However, the resurgence of drug resistance and adverse drug reactions decreases the effectiveness of cART in the treatment of HIV/AIDS [4]. The significance of the NNRTI class of drug is its high potency, selectivity, no intracellular metabolism, and lower cytotoxicity, in comparison to NRTIs [5]. In the last two decades, more than 60 diverse chemical scaffolds have been identified and reported as NNRTIs. Six clinical NNRTI drugs, Nevirapine (NVP), Delavirdine (DLV), Efavirenz (EFV), Etravirine (ETR), Rilpivirine (RPV), and Doravirine (DOR), have been approved by the US FDA against HIV-1 infection. The binding mode of these structurally diverse NNRTIs is similar to a “butterfly”, “horseshoe”, or “U” mode conformation at the hydrophobic pocket of the active site [6]. However, high-level resistance-associated mutations such as K103N, Y181C, L100I, E138K, Y188L, G190A/S, and K101E/P/Q, restrict the clinical usage of most of the current NNRTI drugs. In addition to resistance issues, the first-generation NNRTIs, NVP, and EFV, produce adverse drug reactions such as skin rashes, central nervous system toxicity, liver toxicity, and liver failure. In addition, DLV elicits severe skin rashes due to its high medication load, as much as three times daily in the clinical treatment of AIDS. The dose–dependent prolonged QT interval is observed at doses 75 and 300 mg in the Phase IIB trials of RPV. ETR showed drug–drug interaction effects due to its inhibiting properties against the metabolizing enzymes CYP2C9 and CYP2C19. Furthermore, both drugs, ETR and RPV, have poor water solubility and hence poor pharmacokinetic (PK) properties [4]. Therefore, the design and development of new chemotypes of NNRTIs are required to improve their safety, pharmacokinetic properties, and efficient activity against drug-resistant strains. Medicinal chemistry researchers apply different optimization strategies to identify and develop novel NNRTIs with high potency against wild-type (WT) and mutant strains [7,8]. The formulation experts design various delivery systems for NNRTIs to improve pharmacokinetic profile and sustain therapeutic effect [9]. This review focuses on the identification of emerging novel scaffolds of NNRTIs and various approaches in the fields of medicinal chemistry, pharmacology, and formulation techniques for optimizing the lead molecule as a promising drug candidate.

## 2. Strategies for the Design and Development of NNRTIs

There are several medicinal chemistry-based traditional and modern strategies applied for the identification of NNRTIs effective against the various drug-resistant mutations with improved physicochemical and PK properties. The successful strategies in medicinal chemistry include molecular/fragment hybridization, bioisosterism, scaffold hopping, fragment hopping, conformational restriction, prodrug approach, ligand lipophilic efficiency, high-throughput screening, covalent-binding, targeting highly hydrophobic channel, targeting dual site, and application of multi-target-directed ligand, fragment-, and structure-based approaches. Pharmacological strategies include mono and combination regimens, and the formulation strategy focuses on long-acting formulations that extend the half-life of antiretroviral agents to achieve the desired biodistribution profile. In the subsequent sections, these strategies are described in detail in relation to the developmental process of NNRTI representatives.

### 2.1. Molecular Hybridization

Molecular hybridization is one of the most important and successful strategies applied to the design of NNRTIs. It involves the association of two diverse bioactive pharmacophoric characteristics that form a novel compound with optimal or high potency, selectivity, improved physicochemical properties, and good PK parameters [10]. The novel diarylpyrimidine (DAPY) derivatives are the successful outcome of the molecular hybridization approach with significant improvement in PK and pharmacodynamic (PD) profile. Zhang et al. developed DAPYs with uracil by integrating essential structural components based on the superimposition of the crystal structures of RT complexed with JLJ506 (PDB ID: 4H4O) and ETR (PDB ID: 3MEC). The most active compound **1** demonstrated excellent activity against WT HIV-1 (EC_50_ = 5.6 nM and Selectivity Index, SI > 50,000) and against clinically prevalent mutant strains of E138K with EC_50_ = 34.2 nM (Figure 1). Moreover, it shows moderate potency with EC_50_ = 0.43, 0.086, and 0.15 μM against the common mutant strains of K103N, L100I, and Y181C, respectively. The promising activity of compound **1** is due to the cyanovinyl group, which forms π–π stacking with the side chain of W229. Furthermore, the incorporation of uracil increased water solubility by <0.1 µg/mL [11].

In 2020, another series of DAPYs as NNRTIs were designed with a molecular hybridization approach resulting in the formation of cyanomethyl-linked DAPYs (Figure 1). Among the series of 16 compounds, the most active compound **2** resulted in excellent potency (EC_50_ = 0.027 µM) against WT HIV-1. However, it showed moderate activity against single-point mutations of E138K, Y181C, K103N, and L100I (EC_50_ = 0.17, 0.87, 0.90, and 1.21 µM, respectively). The predicted binding poses of compound **2** using molecular docking revealed that the biphenyl group possibly interacts with the aromatic side chain of Tyr181, Tyr188, Phe227, and Trp229 by π–π stacking and hydrophobic interactions. The cyanomethyl linker forms a direct hydrogen-bond interaction with the amino acid residues Ile180 and Tyr181, and water-mediated interaction with Glu138 located at a distance < 3.5 Å [12].

In another study by Han et al., novel diarylbenzopyrimidines (DABPs) were synthesized with the introduction of the chlorophenyl group of EFV into the diaryl fragment of ETR (Figure 1). Among the compounds synthesized, compound **3** showed potent inhibitory activity against the WT (EC_50_ = 10.6 nM) and EFV-resistant K103N (EC_50_ = 10.2 nM) mutant along with a moderate clearance PK profile (liver microsome clearances of 33.2 μL/min/mg, oral bioavailability estimated as 15.5% in a rat model and 14.4 μL/min/mg liver microsome clearances in a human model). The single-dose toxicity of compound **3** was also determined in rats and no mortality was observed, but it resulted in an abnormal decrease in body weight after administration of a dose of 183 mg/kg body weight. Compound **3** exhibited unsatisfactory activity towards the prevalent E138K mutant strains (EC_50_ = 17.7 nM). Furthermore, structural optimization by removing a chlorine atom from the phenyl ring resulted in the formation of a very potent compound **4** against the variants WT, E138K, and K103N (EC_50_ = 3.4, 4.3, and 3.6 nM, respectively). The cytotoxicity of compound **4** reduced significantly (CC_50_ = 6138 nM) with a higher SI of 1827 and an improved PK profile such as intrinsic microsome clearances of 34.5 μL/min/mg in humans, and improved oral bioavailability (F) to 16.5% at a dose of 5 mg/kg in rats. Compared to compound **3**, single-dose toxicity of compound **4** showed no mortality or observed abnormal reduction in animal body weight at high doses of 293 mg/kg body weight. Docking studies followed by molecular dynamics simulations performed on the E138K or K103N mutated protein revealed that the cyanovinyl group is buried in the hydrophobic tunnel of the protein, forming a hydrophobic interaction. The carbonyl group of Glu138 forms a water-mediated H-bond interaction with the NH linker of the aromatic ring and the 6-nitro group. Additional hydrogen-bond interaction is found between the backbone of Lys101 and the N1 atom of the pyrimidine core [13]. By using the hybridization strategy and additional optimizations including removal of a methyl group from the central pyrimidine ring, and replacement of fluoro with a methyl group on the left ring of difluoropyridinyl-DAPYs, the Li Ding group fused the methyl-pyrimidine-DAPYs and difluoropyrimidine-DAPYs of the known NNRTIs to create 4-pyridinyl-phenyl-DAPY hybrids. Subsequent 4-pyridinyl-phenyl-DAPY hybrid modifications resulted in excellent and enhanced inhibition against wild-type HIV-1 (EC_50_ = 1.7 nM), single mutant viruses (EC_50_ = 257 nM), and double mutant strains (EC_50_ = 64 and 299 nM) [14]. Furthermore, the 4-pyridinyl-phenyl-DAPY hybrids exhibited low cytotoxicity, good solubility, and liver microsome stability. However, the drug’s PK characteristics and inhibitory effectiveness against mutant strains were inadequate. 

### 2.2. Bioisosteric Replacement

Bioisosteric replacement is one of the drug design tools for developing new lead molecules with desirable drug likeliness, physicochemical, and PK properties by applying either classical or non-classical bioisosteric methods. The classical bioisosteric design is dependent on molecular functional groups with the same number of atoms, valence electrons, and degree of unsaturation. The non-classical bioisosteric design is based on molecular fragments with similar pK_a_, electrostatic potential, orbital occupation/HOMOs, and LUMOs [15,16].

In order to overcome HIV drug resistance, Zhan et al. attempted to replace the five-membered heterocycle fragment of triazole and thiadiazole with a six-membered heterocycle, 1,2,4-triazine, leading to a novel scaffold 1,2,4-triazin-6-ylthioacetamides (Figure 2). In this series, compound **5** exhibited stronger potency with an EC_50_ value of 0.018 µM towards the WT HIV-1 and modest inhibition (EC_50_ = 3.3 ± 0.1 µM) against the double mutant strain RES056 (K103N/Y181C). Structure-activity relationship (SAR) and computational analysis showed that the naphthalene ring linked to the 1,2,4-triazine core is essential for activity due to π–π hydrophobic interaction with the aromatic residues Tyr188, Phe227, and Trp229 from the binding site. A crucial hydrogen-bond interaction is observed between the backbone N-H of Lys103 and the amidic carbonyl group of the inhibitor. Moreover, Pro236 residue closely interacts with the 2-bromo-4-sulfonamidephenyl moiety of the inhibitor by electrostatic and Van der Waals interactions enhancing the biological activity [17].

Kang et al. applied the bioisosterism approach by replacing thiophene [3,2-*d*]pyrimidine with piperidine-substituted thiophene[2,3-*d*]pyrimidine. The change in the core ring resulted in a hydrogen-bond interaction with the backbone of the Lys101 residue of the RT enzyme. The binding mode of compound **6** in the mutant strain RES056 showed stronger hydrogen-bond interaction with the main chain of Lys104 and Val106. Even substitutions due to mutation from Lys103 to Asn103 and Tyr181 to Cys181 at the active site of the double mutant RT enzyme do not affect activity [18]. The binding mode analysis of compound **6** in HIV-1 WT RT showed π–π interactions from the left-wing structure towards hydrophobic tunnel residues of Tyr181, Tyr188, Phe227, and Trp229. The analog consists of an amino piperidine-linked sulfonamide structure that forms a hydrogen-bond interaction with the backbone nitrogen of Lys101 and Val106. Compound **6** demonstrated potent anti-HIV-1 inhibitor activity at nanomolar concentration compared to ETR against WT and a wide range of NNRTI-resistant HIV-1 strains (Figure 2). Furthermore, compound **6** exhibited good solubility in water (S = 22.8 μg/mL at pH = 7.0) and a minimal inhibitory effect on CYP isoforms with IC_50_ > 5.0 μM and the hERG channel with an IC_50_ value of 0.979 μM. The PK values of compound **6** indicate insignificant drug–drug interactions and a reduced risk of cardiotoxicity. The determined LD_50_ parameter in mice is greater than 2000 mg/kg and in vivo oral bioavailability resulted in 37.06%. These promising PK profiles encouraged investigation of compound **6** as a potential anti-HIV-1 drug candidate.

Chen et al. reported the structural modifications of chiral carbon with hydroxyl at the para-position of a biphenyl DAPY template using the bioisosterism strategy, resulting in the formation of racemic compounds. The (S)-(−) enantiomer of compound **7** exhibited better inhibitory activity in low nanomolar concentration range against the WT and single mutant strains of L100I, K103 N, Y181C, E138K, and Y188L than the (R)-(+) enantiomer. It showed submicromolar inhibition against the double mutant virus strains of F227L + V106A and RES056 (Figure 2). In addition, the acute toxicity analysis and the PK evaluation of compound **7** in mice do not show behavioral abnormalities or damage to vital organs, and it is also well tolerated at a dose of 2 g/kg. However, compound **7** has poor metabolic stability, resulting in a decrease in oral bioavailability [19].

### 2.3. Scaffold Hopping

The scaffold hopping approach is applied to explore structurally diverse compounds from a series of active compounds by altering the scaffold of the molecule. This approach helps to improve solubility by replacing the lipophilic scaffold with a polar substituent. The PK properties can be improved by replacing the metabolically unstable scaffold with a more stable and non-toxic scaffold. Sometimes, the flexible peptide backbone is replaced with a rigid scaffold to enhance the binding affinity against the target protein. Several candidate molecules were successfully developed by applying the scaffold hopping approach targeting the binding sites of the ligands benzodiazepine and dopamine and the enzyme cyclooxygenase [20,21,22].

In NNRTIs, the successful development of scaffold replacement of molecules includes cycloalkyl arylpyrimidines (CAPY), biphenyl-DAPY, thiophene-biphenyl-DAPY, DABPs, sulfinylacetamide-DABPs, and hybrid ETR-VRX-480773 (Figure 3), which showed significantly superior activity against HIV-1, particularly against mutant strains [13,23,24,25,26,27,28,29].

Recently, Li et al. developed a new derivative of CH(CN)-DAPYs from CH(CN)-DABOs by employing the scaffold hopping strategy (Figure 3). The cyano-methylene linker of CH(CN)-DABOs is introduced onto ETR with addition of a methyl group to the central pyrimidine ring to improve the non-polar contacts at the binding pocket of RT. Compounds **8** and **9** showed good potency against WT HIV-1 (EC_50_ = 6 and 8 nM, respectively) and K103N mutant (EC_50_ = 60 and 80 nM) in nanomolar concentration [30]. By using the scaffold hopping and molecular hybridization techniques, Wang et al. transformed the chemical structures of previously identified thiophene[3,2-d]pyrimidine derivatives into dihydrothiopyrano[4,3-d]pyrimidine derivatives. Notably, as compared to ETR and RPV, compound **9a** showed very powerful antiviral activity (EC_50_ = 4.44–54.5 nM) against a variety of HIV-1 strains, decreased cytotoxicity (CC_50_ = 284 µM), and enhanced resistance profiles (RF = 0.5–5.6). Furthermore, **9a** demonstrated superior solubility (12.8 g/mL), no discernible inhibition of primary CYP enzymes, and notable metabolic stability and in vivo safety characteristics, and the hERG inhibition profile of **9a** (IC_50_ = 19.84 µM) showed a notable improvement [31].

### 2.4. Conformational Restriction

The conformational restriction (rigidification) design strategy is applied to the bioactive flexible ligands to prevent entropic loss. In particular, the restriction of the rotatable bond by introducing a fused-ring structure is designed to maintain a favorable binding conformation and orientation of the compound in the binding pocket of the target protein to achieve better biological potency, selectivity, and reduced metabolic deactivation.

The NNRTIs bind in the form of a butterfly-like or horseshoe-like conformation at the allosteric site to disrupt the RT polymerization function. The spatial arrangement of NNRTIs is mainly due to the forces of the hydrophobic group, H-bond donor, and acceptor between the NNRTI (chemical scaffold groups) and the tunnel (lipophilic and polar residues) in the active site. Therefore, it is possible to examine the limited intramolecular factors involved in maintaining the preferential butterfly-like or horseshoe conformation. Three rigidification factors such as rigid heterocycles located in the ‘body-linker’, intramolecular H-bond, and stereochemical diversity-oriented restrictions were reported for the conformational restriction of NNRTIs [32].

The rigid heterocyclic scaffolds were incorporated in NNRTIs as conformation-restricted motifs to improve anti-HIV potency in WT and mutant strains including fused tricyclic platform in R82913 (**10**), pyrimidine in TMC278 (**11**), indole in IDX-899 (**12**), triazole in RDEA-806 (**13**), 2-oxopyrrolidine in *N*-aryl pyrrolidinone (**14**)**,** and indazole in **15**. The promising potency of the phenylethylthiazolylthiourea trovirdine (LY 300046·HCl; **16**), MIV-150 (**17**), and benzimidazolone (**18**) is achieved by conformationally restricted factors with their intramolecular hydrogen bonds (Figure 4A) [33,34,35,36,37,38,39,40].

The integration of the stereocenter played a vital role in modifying the NNRTIs’ conformational behavior as potential drug candidates with improved effectiveness for anti-HIV activity and prospective utility in controlling drug-induced mutations. Numerous NNRTIs possess chiral cyclopropane functional groups to maintain conformational rigidity. The analogs presented in Figure 4B, with a stereocenter cyclopropane ring, possess nanomolar inhibition toward several clinically significant mutants: urea-phenethylthiazolylthiourea analogs (**19**–**22**), tetrahydroquinoline (**23**), *S*-dihydroalkoxybenzyloxopyrimidine (DABO) (**24**), oxindole (**25**, **26**), quinolones (27), and cyclopropyl indole (**28**) [40,41,42,43,44,45,46,47].

The DABO derivative compounds **29** and **30** contain two methyl groups, one at the benzylic carbon and other at the fifth position of the pyrimidine ring without compromising the conformation (R enantiomer) of the compounds, resulting in a snug fit into the NNRTI binding pocket. F2-S-DABO derivative **29** exhibited more potent HIV-1 inhibition than the reference compound MKC-442 and NVP. The F2-NH-DABO derivative **30** has remarkable activity against the Y181C variant. Some of the stereocenter derivatives, namely the R enantiomer of N,N-DABO **31** (MC1501) and DABO-DAPY **32** (diarylpyrimidines, MC2082), possess higher HIV inhibition than the respective S enantiomer and racemic mixtures [48,49]. The binding affinities of 3-arylphosphoindole **33** and CH(OH)-DAPY **34** to RT and their biological potency are completely based on the presence of a chiral center. The R enantiomer is highly potent in comparison to its S enantiomer (Figure 4B) [50,51].

Recently, in 2019, Sang et al. applied the conformational restriction strategy to thiophene-biphenyl-DAPY derivatives by introducing fluoro, chloro, or methyl groups on the biphenyl ring system. Especially compounds **35** and **36,** possessing two methyl groups, displayed potent anti-HIV activity against the WT (EC_50_ = 14 and 17 nM, respectively), and a panel of clinically relevant HIV-1 mutant strains of L100I, K103N, Y181C, and E138K (EC_50_ = 0.13, 0.02, 0.03, 0.04 and 1.97, 0.13, 195.72, 0.16 µM, respectively) compared to the reference drug, ETR. The potent activity of compounds **35** and **36** is mainly due to the methyl groups substituted on the biphenyl ring that improve the theoretical dihedral angle on the biphenyl ring, resulting in influential face-to-face π–π interactions with Y181/Y188 and interactions with the F227/L234 region of the hydrophobic pocket (Figure 4C). This dihedral angle contribution possibly makes the conformation of the biphenyl group perfectly fit into the NNRTI active site. Furthermore, the appropriate balance of flexibility and the obligatory restriction in the structure of NNRTIs for the formation of bioactive conformations will resolve the prospect of promising bioactive molecules against HIV-1-resistant mutants [27].

### 2.5. Prodrug

The prodrug strategy is a synthetic approach in which three components, a parent drug, a spacer, and a specifier, are conjugated together, forming a pharmacologically inactive compound. The inactive prodrug compound transforms into an active compound along with a specifier by enzymatic cleavage at the spacer part of the prodrug. It is normally designed to solve the parent drug or compound PD and PK problems associated with metabolic instability, toxicity, lack of specificity, and solubility issues [52,53].

Petersen et al. reported the first double-prodrug strategy applied to a NNRTI, MKC-442, and NRTI, d4T, by incorporating the SATE (*S*-acyl-2-thioethyl) group (Figure 5A). Initially, monophosphate NRTI d4T and the N-3 position of NNRTI 6-Benzyl-1-(ethoxymethyl)-5-isopropyluracil (MKC-442, Emivirine) were masked and protected with SATE and *p*-hydroxybenzoyl groups, respectively, before synthesizing the double-prodrug **37**. Finally, both NRTI and NNRTI compounds were bound to the N-3 position of MKC-442 through a labile *p*-hydroxybenzoyl protection group. Furthermore, two more dual-prodrugs **38** and **39** were prepared using the d4T structural portion connected by two N-1 substituted phenols, **40** and **41,** as precursors of a NNRTI. There are two possible routes of degradation mechanism for the double-prodrug **37** outside and inside of the cell. In *Route A*: first MKC-442 is cleaved off from compound **37** and prodrug **37a** enters into the cell. In *Route B*: a hydrolytic reaction takes place inside the cell to release the active components of d4T monophosphate and MKC-442. The double-prodrug **37** is transported from outside to inside the cell and then hydrolyzed into the active component. Stability studies proved that both double-prodrugs **38** and **39** released the d4T monophosphate and the NNRTIs **40** or **41** after the compounds **38** and **39** were transported inside the cell. All three synthesized double-prodrugs **37**, **38**, and **39** proved to be potent inhibitors against the WT HIV-1 and Y181C mutant strains in both CEM WT and CEM TK- cell lines [54].

RDEA427 is a DAPY analog-based NNRTI entering into the preliminary clinical evaluation as a drug candidate due to its wide range of mutant HIV-1 strain inhibition in nanomolar concentration. Unfortunately, the clinical studies of RDEA427 were discontinued in 2010 because of undisclosed reasons which were speculated to be inappropriate physicochemical properties [55]. Thus, Huang et al. developed the first carbonate prodrug of NNRTI drug candidate RDEA427 by treating with chloromethyl isopropyl carbonate in the presence of sodium hydride in tetrahydrofuran at room temperature (Figure 5B). The carbonate prodrug **42** showed potent in vitro activity (EC_50_ = 0.0055 µM) for the WT HIV-1 strain when compared to the parent drug RDEA427 WT activity (EC_50_ = 0.0029 µM), but against K103N/Y181C (RES056) strains, compound **42** showed moderate inhibition with an EC_50_ of 0.15 µM. Metabolic stability test analysis in human plasma confirms the presence of RDEA427 released by **42** (carbonate prodrug) during the initial 1 h in a linearly time-independent behavior [55].

Wang and his co-workers prepared the carbonate and phosphate prodrug of DOR by treating chloromethyl isopropyl carbonate or di-*tert*-butyl(chloromethyl)phosphate reagent with DOR in dimethyl formamide solvent, yielding the target compound **43** along with a BOC-protected intermediate, di-*tert*-butyl ((3-((3-(3-chloro-5-cyanophenoxy)-2-oxo-4-(trifluoromethyl) pyridin-1(2H)-yl)methyl)-4-methyl-5-oxo-4,5-dihydro-1H-1,2,4-triazol-1-yl)methyl) phosphate. Subsequently, deprotecting the intermediate at 40 °C in acetone/water solvent yielded compound **44**. Unfortunately, the two prodrugs **43** and **44** showed inferior anti-HIV-1 activity compared to the parent compound, DOR. In terms of solubility analysis, carbonate prodrug **43** had similar aqueous solubility and the phosphate prodrug had improved aqueous solubility as compared to DOR. This result demonstrated that the phosphate group provide more solubility in aqueous media than the carbonate group in DOR. The prodrug **44** did not experience any degradation when tested in PBS buffer at pH 2.0 and 7.4 after 24 h duration and was chemically stable (Figure 5B). Moreover, another strategy of molecular hybridization and bioisosterism to improve the potency of the DOR scaffold was attempted by maintaining the left cyanochlorophenol moiety and pyridine central ring of DOR and the acetamide linker from compound **45** (GW-678248), resulting in the development of novel acetamide-substituted DOR analogs. In this series, two compounds **46** (EC_50_ = 59.5 nM) and **47** (EC_50_ = 54.8 nM) exhibited promising activity against HIV-1 WT strain and modest activity against the double RT mutant (K103N+Y181C) HIV-1 RES056 strain, with EC_50_ = 0.343 and 0.196 μM for compounds **46** and **47**, respectively [56].

### 2.6. Ligand-Lipophilicity Efficiency (LLE)

Ligand-lipophilicity efficiency (LLE) optimizing strategies were applied for the successful finding of novel NNRTI chemotypes to improve ligand bioactivity against HIV and metabolic stability with reduced clearance and increased half-life (t_1/2_) [57,58]. For instance, optimizing the heterocyclic pyrrole scaffold in compound **48** (Capravirine) with a pyrazole ring along with modification of pyridine (high lipophilic) with the ethyl (low lipophilic) moiety led to the development of compound **49** (UK-453061, Lersivirine). This refinement resulted in improved potency and wide-spectrum activity against clinically relevant RT mutations (Figure 6) [59,60,61]. Clinical studies were initiated for compound **49** due to its significant improvement in LLE compared to compound **48**, and metabolic stability (t_1/2_ = 73 min). Currently, LLE is being used as a drug discovery tool or applied as a filter for the compounds obtained from the structure-based virtual screening approach [62,63,64,65,66].

### 2.7. Structure-Based Optimization

Currently, structure-guided strategy is one of the widely employed approaches for designing new chemotypes of NNRTI by preserving structural flexibility inside the non-catalytic site of RT that address the resistant problems such as loss of hydrophobic interactions (Y181C, Y188L, and F227L), steric hindrance (L100I and G190A/S), and pocket entrance mutations (K101E and K103N). Based on the crystal structure of RT in complex with NVP, a large number of structure-based approaches were proposed in the design of novel NNRTIs against the RT allosteric site to solve the major issue of resistance-associated mutations [67]. The discovery of DAPY inhibitor-derived drugs such as ETV and RPV possessing conformational flexibility preserves efficacy against a wide range of resistant variants. The flexible linker between the center of the pyrimidine ring and the diaryl ring resulted in a unique torsional angle that increases the overall stability and is adaptive to the binding pocket of resistant strains of the HIV-1 RT enzyme. Based on this successful discovery, new pharmacophore-based analogs similar to DAPYs were designed and developed [68].

For instance, compounds such as catechol diether (**50**), IDX899 (**51**)**,** and piperidine-substituted thiophene pyrimidines (**52**) were designed with the support of the crystal structure of the RT enzyme. Compound **50,** having a CN-indolizine ring and terminal ether-linked uracil, exhibited potent nanomolar activity for several RT variants of WT and mutant strains of Y181C, K103N, and K101P. This was achieved by optimizing to a horse-shoe conformation in the crystal structures of RT (WT) and several RT variants and facilitating numerous interactions within the entrance, groove, and tunnel of NNIBP (Figure 7) [69,70,71].

Another series of aryl-phospho-indole scaffolds as flexible NNRTIs was developed by introducing a phosphinate linker to EFV. Among them, compound **51** (IDX899) had promising activity against single mutant K103N and double mutant Y181C/K103N variants along with superior aqueous solubility to 20 μM, i.e., >40-fold step up compared with EFV aqueous solubility (Figure 7). Three vital interactions were observed in the crystal structures of RT (Y181C/K103N) and IDX899, showing an H-bond with Lys101 residue, π–π interactions with Pro236 and Tyr188 residues, and Van der Waals interaction with Phe227, Val106, Asn103, and Leu100 residues [72].

Chan et al. developed compound **53** consisting of reactive α-chloro ketone and acrylamide linked to a naphthyl ring that act as ‘electrophilic warheads’ interacting covalently to the thiol group of Cys181 residue in RT (Y181C) and RT (K103N/Y181C) variants (Figure 7). This compound exhibited time-dependent activity with nanomolar to micromolar activity in in vitro cell-based anti-HIV and enzymatic assay, respectively. Based on the available crystal structures of RT and docking methods, a rationale to identify new NNRTIs was developed with better therapeutic efficacy and PK profile [73].

Using the molecular docking and molecular dynamics methods, ETV derivatives were designed by incorporating the hydrophilic fragments of morpholine, methyl sulfonyl, and sulfamide piperazine/piperidine heterocycles in the solvent-exposed areas of the NNIBP, and compound **54** was identified, possessing high water-solubility (30.92 μg/mL) with favorable PK properties [74,75].

With the support of computational and structure-based design strategy, a long-acting late-stage preclinical candidate **55** (naphthyl catechol phenyl ether) was developed, belonging to the class of catechol diether, which shows nanomolar potency (EC_50_ = 1.9 nM) against the WT and clinically significant HIV-1 drug-resistant strains, non-cytotoxicity, and non-blockade effects on CYP3A4 enzymes and the hERG ion channel. To synthesize compound **55**, several optimization steps from the lead compound were involved for transforming the hit molecule identified from in silico approach with micromolar to sub-nanomolar activity, such as structural modification of diphenylmethane which includes appropriate functionalization, changing the scaffold to diphenyl ether, and alteration of substitutions. Compound **55** exhibited synergistic antiviral effects in combination with FDA-approved drugs (FTC, TDF, and EVG) in HIV-1–infected MT-2. The nano-formulation of compound **55** using poly(lactide-coglycolide) acid (PLGA) released the compound continuously for ∼3 weeks and maintained the efficacy after a single dose (Figure 7) [76]. Based on the X-ray crystallographic analysis and previous SAR studies of three pivotal pharmacophores at the DAPY derivatives, Huang et al. designed and synthesized two series with six-membered cyclohexyl and five-membered pyrrole rings fused at the central ring B of DAPYs as pyridyl-bearing fused bicyclic derivatives **55a** which possessed potent antiretroviral activity against the wild-type (WT) HIV-1 strain or multiple NNRTI-resistant strains at low nanomolar levels and improved metabolic stability profiles with lower intrinsic clearance compared to RPV [77].

In order to investigate the chemical modifications around the entrance channel of the HIV-1 RT binding pocket, Xu et al. designed and synthesized a series of novel indolylarylsulfones (IASs) bearing phenylboronic acid and phenylboronate ester functionalities at the indole-2-carboxamide. At the cellular level, the antiviral findings demonstrated potent inhibitory effects against WT HIV-1 (EC_50_ (IIIB) = 6.7–42.6 nM) by all the derivatives. In particular, the representative compound **55b** (EC_50_ (IIIB) = 8.5 nM) maintained its inhibitory activity against the mutant strains L100I, K103N, and V106A/F227L (EC_50_ (L100I) = 7.3 nM, K103N = 9.2 nM, and F227L/V106A = 21.0 nM), far exceeding the effectiveness of the commercially available drugs NVP, 3TC, and EFV and equivalent to ETV [78].

### 2.8. Fragment Hopping

Currently, the fragment hopping approach is seen as an alternative to conventional drug discovery methods and is gaining interest in the discovery of anti-HIV drugs. Interestingly, recent progress in ligand/structural methodologies for developing drugs in combination with the fragment hopping approach has developed robust tools for the development of antiretroviral agents. Han et al. adopted this approach to develop a new sulfinyl acetamide DAPY as a NNRTI by using the thioacetamide fragment present in the most potent NNRTIs of VRX-480773, RDEA806, and dihydro-alkylthio-benzyl-oxopyrimidines (S-DABOs). After the insertion of a thioacetamide fragment in the DAPY scaffold, further oxidation yielded the target series of sulfinylacetamide-DAPYs by modifying thioethers to sulfoxides. The compound **57,** containing sulfoxide with a 4-CN group on the phenyl ring, showed ~12 and 20-fold improved anti-HIV activity against WT and K103N mutant strains compared to the thioether series compound **56**. Interestingly, the most active compound **57** exhibited improved antiviral activity (EC_50_ = 0.0249 µM against the HIV-1 WT strain and an EC_50_ = 0.0104 µM against the K103N mutant strain), low cytotoxicity (CC_50_ > 221 µM), and a high selectivity index (SI WT > 8873, SI K103N > 21,186). The cytotoxicity of compound **57** is lower in comparison to ETR and RPV (Figure 8). The PK properties of compound **57** revealed inherent liver microsome stability in human (8.45 mL/min/mg) as well as in rat (8.58 mL/min/mg) models. The half-life of the compound **57** showed higher in human (t_1/2_ = 82.0 min) than in the rat (t_1/2_ = 80.8 min) models. The docking analysis of compound **57** with mutant K103N RT showed that the improvement in antiviral activity was due to the development of new hydrogen-bond and stronger van der Waals interactions [29]. The same research group further developed another two series of sulfur-containing diarylbenzopyrimidines as NNRTIs, applying the fragment hopping strategy by combining thioacetamide with previously disclosed compounds, DABPs, and subsequent oxidation of sulfur at thioacetamide fragment. The most potent compound **58** displayed comparable potency against the K103N mutant virus (EC_50_ = 0.0228 µmol/L) compared to ETR (EC_50_ = 0.0037 µmol/L), and also showed higher potency than NVP and EFV (EC_50_ = >0.11 and 0.124 µmol/L) (Figure 8). However, compound **58** exhibited a remarkable decrease in cytotoxicity (CC_50_) with a value of 99.6 µmol/L compared to the standard NNRTIs (NVP = >15.0 µmol/L, EFV = >6.34 µmol/L, >ETR = 4.59 µmol/L, and RPV = 4.38 µmol/L). Apart from the cytotoxicity, compound **58** has higher water solubility of 300 ng/mL at pH 7 than RPV (20 ng/mL) and the parent compound of DABP (undetectable at 20 ng/mL) [79]. Ding et al. applied a fragment-based replacement technique to substitute various heteroaromatic-biphenyl-DAPYs on the scaffold of biphenyl-DAPYs in order to improve druggability. These molecules show good HIV-1 inhibitory activity at nanomolar concentrations. Among these, **58a** with a 4-pyridine group demonstrated good inhibition of both the WT and mutant HIV viruses with notable selectivity. Furthermore, compared to ETR and RPV, **58a** molecule showed a respectable improvement in druggability: (1) In several pH ranges, **58a** hydrochloric acid salt showed noticeably enhanced water solubility. (2) Neither acute toxicity nor significant CYP enzymatic inhibitory action was present in **58a**. (3) Excellent oral bioavailability of **58a** (F = 126%, in rats) [80].

### 2.9. High-Throughput Screening

Several cell lines and assays have been established successfully for cell-based high-throughput screening (HTS) to identify novel scaffolds as HIV-1 NNRTIs. Wang et al. identified a novel sulfanyltriazole as a NNRTI through HTS using a cell-based assay, and further structural optimization of the sulfanyltriazole scaffold was performed by substituting various groups on the carboxamide phenyl ring and the other phenyl rings attached to the triazole moiety. Among the two series, compound **59**, with *o*-chloro (EC_50_ = 0.003 (WT), 0.023 (Y181C), 0.065 (K103N), and 0.182 µM (K103N/L100I)), and compound **60**, with *o*-nitro substitution (EC_50_ = 0.001 (WT), 0.016 (Y181C), 0.006 (K103N), and 0.024 µM (K103N/L100I)) on the carboxamide phenyl ring, showed promising activity against the Y181C, K103N, and double mutant K103N/L100I mutant strains (Figure 9) [81].

Elinder et al. screened a library of 800 NNRTIs against four clinically relevant variants of HIV-1 RT (WT, L100I, Y181C, and K103N) by using a surface plasmon resonance-based biosensor method. Compound **61**, containing a chromane nucleus and thiourea group, was identified as a promising inhibitor with high affinity towards the four resistant HIV-1 RT strains when compared to NVP and EFV (Figure 9) [82]. Elleder et al.’s research group performed HTS and identified 3-aminoimidazo[1,2-a]pyridine scaffolds as HIV-1 RT inhibitors from an in-house diverse set of synthetic compound collections at the Salk Institute by using a plate-based assay containing VSVg-pseudotyped HIV-1 vector (pNL4-3LucR+E-) encoding firefly luciferase. The imidazo[1,2-a]pyridine pharmacophore-containing compound **62** has revealed activity against wild-type and some of the common drug-resistant variants of K103N, V106A, Y188L, and Y181C HIV-1 reverse transcriptase (Figure 9). However, this new chemical scaffold lead of **62** is structurally isosteric with the benzimidazole-based potent NNRTI compound **63**. Compound **62** is a suitable NNRTI lead for further development as an antiviral agent, due to its simple synthetic route and inexpensive, single-chemical transformation reaction compared to the synthesis of compound **63** [83]. A novel potent NNRTI belonging to the chemical class of aryl-substituted triazine was identified by Kim’s group from an HTS campaign of 200,000 compounds for anti-HIV activity using a cell-based full replication assay. The identified hit of triazine scaffold-based compound **64** exhibited nanomolar inhibitory activity (EC_50_ = 265 nM) against HIV replication and also showed moderate potency in an in vitro RT polymerase assay. Further structural modifications of the parent compound **64**, with the triazine core replaced with pyridine, resulted in the formation of phenylaminopyridine scaffold-containing compounds. The substitution of a chloro group in the pyrazole moiety (compound **65**) resulted in excellent antiviral activity against wild-type (WT) (EC_50_ = 0.02 nM) and also in the key RT mutations of Y181C (EC_50_ = 16 nM) and K103N (EC_50_ = 7.3 nM) (Figure 9). Moreover, compound **65** did not exhibit any inhibitory or induced effects on five isoforms of the cytochrome P450 (CYP) and CYP 3A4 enzymes, respectively, at concentrations given up to 10 μM. Physicochemical properties such as solubility profile (1.1/132 µM) also increased due to the presence of substituted pyrazoles with a polar moiety and also due to the formation of hydrochloride salt. Notably, the PK profile of compound **65** showed bioavailability above 70%, improved AUC value (67 µM/h), moderate clearance rates (CL = 8.2 mL/min/kg), and half-life (t_1/2_ = 2.0 h) [84]. The high-performance HTS strategy and exploration of internal compound collections have been effective in identifying and developing many NNRTIs; however, there are several drawbacks, including time-consuming systemic improvements, very low hit rates, and a high number of false positives.

### 2.10. Covalent Bonding Inhibition

Covalent inhibitors, which offer pharmacological benefits including reduced dosages and longer duration of action, can covalently connect to the target proteins by unique orientation and localization of covalent warheads. More significantly, covalent binding can boost small compounds’ affinity for their targets, enhancing resistance [85,86]. The Tyr181 Cys mutation of the residue Tyr182 in HIV-1 RT has resulted in resistance and impedes the development of NNRTIs. It is the main point mutation resulting from the consumption of NVP, the first-in-class medication that is still frequently used, particularly in poor nations. By using in vitro cell experiments, mass spectrometry, and protein crystallography, Chan et al. discovered compounds **66** and **67** to be covalent inhibitors of both Y181C and K103N/Y181C HIV-1 RT. For both compounds interacting with Cys181, the inhibitor activities of **66** and **67** were increased in potency (lower IC_50_ values) with increasing time. In these situations, an increasing amount of the enzyme is inactivated over time. Additionally, after 48 h, the variants’ IC_50_s for **66** and **67** were noticeably improved (0.14–0.19 µM) in comparison to their WT (IC_50_s 1 µM), demonstrating a 10-fold increase in potency. To prevent off-target effects, two electrophilic warhead types with relatively low activity, such as halo amides and acrylamides, were chosen and incorporated using compound N-(6-cyano-3-(2-(2-(2,4-dioxo-3,4-dihydropyrimidin-1(2H)-yl)ethoxy)phenoxy)-4-methylnaphthalen-1-yl)-2-fluoro-N-methylacetamide as the lead. The thiol group of Cys181 interacts with the moderately active warheads. The structures for the complexes with Y181C RT in Figure 10 include the C-S single bonds between **66** and **67**, as well as Cys181 [73].

Gao et al. presented a new series of indolylarylsulfones having acrylamide or ethylene sulfonamide reactive groups as warheads. In nucleotide incorporation inhibition experiments, compounds **68** and **69** showed better selectivity towards the Y181C mutant than against the wild-type RT. Additionally, the mass shift of the p66 subunit at a low concentration of **68** promoted the covalent alteration of Y181C RT (Figure 11A). However, when the quantity of **69** was raised, additional cysteine residues in RT were also altered, and these off-target effects plausibly explained its significant cytotoxicity [87].

Ippolito et al. reported covalent inhibitors of wild-type HIV-1 reverse transcriptase (CRTIs). Three compounds developed from catechol diether as NNRTIs, possessing a fluorosulfate warhead, have been shown to covalently alter HIV-RT Tyr181. All of them covalently change Tyr181 to generate biaryl sulfate between the tyrosine oxygen atom and the warhead’s sulfur. The sulfate and uracil groups’ oxygen atoms were linked in an extensive hydrogen-bond network with the backbone nitrogen of Lys101 and Lys103. The three analogs of CRTIs, **70**, **71**, and **72**, exhibited strong in vitro suppression against WT RT and cytopathic protection of HIV-1-infected human T-cells with IC_50_ and EC_50_ values in the range of 5–320 nM (Figure 10 and Figure 11B). The covalent modification of Tyr181 by **70**, **71**, and **72** was reflected in an increase in mass of intact protein determined using electrospray ionization–time-of-flight mass spectrometry (ESI-TOF MS) [88].

Zhou et al. abandoned the hERG-perturbing piperidine-linked benzenesulfonamide scaffold of DAPYs, kept the benzene right wing of ETR and inserted an oxysulfonyl fluoride covalent warhead to make a covalent interaction with the highly conserved Tyr318. The target inhibitor, 3-((4-(4-cyano-2,6-dimethylphenoxy)thieno[3,2-d]pyrimidin-2-yl)amino)phenyl sulfurofluoridate, which has a fluorosulfate warhead, was reported to be a strong inhibitor (EC_50_ = 11–246 nM) against HIV-1 IIIB and a panel of NNRTI-resistant strains, considerably outperforming NVP and EFV, and displayed decreased cytotoxicity (CC_50_ = 125 µM). The compound had an IC_50_ value of 0.057 µM in the reverse transcriptase inhibitory experiment using the ELISA technique, and the MALDI-TOF MS data indicated ZA-2’s covalent binding mode with the enzyme [89].

### 2.11. Targeting Highly Hydrophobic Channels

Maximizing highly conserved site contacts, particularly to extend additional interactions with highly conserved hydrophobic residues, has been a sensible approach to counteract drug resistance. Therefore, it has been predicted to be an appropriate strategy to design inhibitors that exclusively interact with highly conserved residues while lowering dependency on potentially variable residues for treating continually arising mutants [90,91].

The common cyanovinyl moiety of RPV and piperidine-substituted thiophene[3,2-*d*]pyrimidine spreads into the hydrophobic channel formed by Tyr181, Tyr188, Phe227, and Trp229, based on the analysis of the co-crystal structures of the two proteins with HIV-1 RT. This interaction with the nearby highly conserved residues Phe227 and Trp229 helps to improve antiresistance profiles. However, it is possible that the electrophilic cyanovinyl group responsible for the significant cytotoxicity acts as a “Michael acceptor”, leading to off-target covalent alteration of proteins, nucleic acids, or other biological components. For this reason, Zhan and his coworkers originally substituted aromatic rings or aliphatic groups connected by the alkyne or alkene to fulfill the spatial orientation, which resulted in two interesting compounds **73** and **74** featuring ethynylpyridine and ethynylethanol in the left wing, respectively (Figure 12). Due to the stability, excellent solubility, and hydrogen-bonding capabilities of the triazole ring, a series of 1,2,3-triazole derivatives were synthesized through the quick and effective CuAAC “click reaction” in order to further leverage the SARs of the hydrophobic channel (Figure 12). The meta-methylbenzoate analog **75** and the para-methylbenzoate analog **76** of the series both displayed nanomolar activity against WT and K103N mutant strains with high selectivity indices and decreased cytotoxicity [92].

### 2.12. Targeting the Hydrophilic Solvent-Exposed Region

Within or near the ligand-binding sites of the target proteins, the solvent-exposed regions are potential binding sites and possibly accommodate structurally diverse moieties and form additional interactions providing broad chemical space for substantial modifications. The addition of solubilizing substituents into the solvent-exposed region has proved equally efficient for enhancing the solubility-limited physicochemical features along with the widely utilized prodrug technique.

Due to its outstanding physicochemical, biological, and easily accessible qualities, morpholine is used as a solvent-friendly fragment in several approved drugs or bioactive compounds. There are enough data to conclude that morpholine not only has a variety of therapeutic effects but also possesses the ability to alter drug-like characteristics. Recent developments include the rational design of a number of new DAPY derivatives with favored morpholine in the piperidin-4-yl-amino and different aromatic or nonaromatic fused pyrimidine rings in the central scaffold. As anticipated, four compounds (**77**–**80**) showed dramatically enhanced solubility while retaining mild antiviral activity (Figure 13). The morpholine moiety was strategically positioned in the solvent-exposed region of the binding pocket and enhanced the solubility of the compounds [74,93].

A hydrophilic methylsulfonyl-substituted piperazine moiety was added to the morpholine-containing compound **81** to increase anti-HIV-1 potency by maintaining good solubility. The modified compound **82** has superior activity compared to ETV against WT and E138K HIV-1 strains and low cytotoxicity. Furthermore, **82** demonstrated improved solubility and a lack of apparent CYP isozyme inhibitory action as well as acute or subacute toxicity. Focusing on the substituted piperazine fragment pointing towards the tolerant region I of NNIBP, the unification of the dominant scaffolds of piperidine-substituted thiophene[3,2-*d*]pyrimidine and **83** resulted in the discovery of thiophene[3,2-d]pyrimidine **84** and thiophene[2,3-d]pyrimidine **85** (Figure 13) which feature the sulfonamide group with substantial increase in solubility without loss of potency [94]. The metric “Fraction of sp3 carbon atoms” (Fsp3, the number of sp3 hybridized carbons/total carbon count) has been developed to assess the complexity of molecular space structure and gauge the carbon saturation of molecules. The introduction of various saturated polar groups to enrich the SAR around the tolerant region I was directed by this advanced design assumption leading to the discovery of compounds with a thiomorpholine-1,1-dioxide moiety, which exhibited improved water solubility and profile towards drug resistance against a number of single mutant strains [95].

### 2.13. Targeting Dual Sites

With the emergence of designing multiple ligands in medicinal chemistry as a strategy to improve therapeutic efficacy and decrease adverse drug responses, the significance of developing dual-acting enzyme inhibitors has increased. RT-dependent ribonuclease H (RNase H) and integrase (IN) play crucial roles in stable infection, hence these enzymes are interesting targets for drug development. RNase H inhibitors have received minimal attention, and only drugs that target the RT polymerase and IN strand transfer activities have been approved for therapeutic application. The active sites of IN and RNase H are highly similar, composed of five-stranded sheets encircled by helices that share the same metal-coordinating DDE amino acid triad necessary for catalytic activity. Furthermore, both employ comparable divalent metal cofactors in their catalytic activity. The five strands of the IN sheets and the equivalent five strands of RNase H sheets are tightly superimposed. However, several distinct characteristics were identified; for instance, IN α1 is separated from the four sheets by around 6 Å compared to RNase H α1. RNase H helix α2 is a three-turn helix whereas IN helix α2 is a one-turn helix. IN has two extra helices that are missing in RNase H. Residues 171 to 186 in IN α5 are next to and parallel to residues β3, whereas residues 196 to 208 in IN 6 are on the same side as residue α1 and are oriented at about a 90-degree angle to residue α5. The relatively similar pharmacophores for the two enzymes justify their use as prospective dual-target inhibitors. The Mg^2+^ ions are completely neutralized by the four highly conserved amino acid residues D443, E478, D498, and D549, which make the RNase H site electrostatically neutral. However, only three acidic residues D64, D116, and E152 oppose the positive charges carried by the Mg^2+^ ions in the IN active site. This makes it possible for the acidic molecules containing a negatively charged carboxylate moiety to interact preferentially with HIV-1 IN rather than RNase H. Based on this analysis, several compounds have the potential to be effective inhibitors of both enzymes [96].

One of the earliest studies on the design and synthesis of IN inhibitors was in 2000 when Huzada and his coworkers synthesized a number of aryl diketo acid (DKA) analogs. Di Santo et al. investigated the potential dual inhibitory effects of several pyrrolyl and quinolonyl DKA analogs against both enzymes [97]. More research on the compounds from these two families was subsequently performed in order to develop dual RNase H/IN inhibitors. As RT-associated RNase H/IN dual inhibitors, pyrrolyl DKA analogs were first presented by Costi and his coworkers in 2013 (Figure 14). Compound **86** was shown to be the most powerful inhibitor, with IC_50_ values for RNase H and IN enzymes of 2.5 µM and 26 nM, respectively [98]. It is interesting to note that esterified analogs of compound **86** showed a small increase in HIV-1 RNase H selectivity compared to IN, although acid variants were more potent against IN. Ester functionality was more suited for dual targeting of IN and RNase H, possibly as a result of the distinct electrostatic characteristics of the active sites of the two enzymes that were previously described. These findings resulted in understanding an effective SAR analysis. In conclusion, compounds **87** and **88** showed the highest inhibitory efficacy when the DKA chain and a phenyl moiety were positioned at 3 and 4, respectively. Additionally, SAR analysis indicated that the inhibitory activity of both enzymes decreased when the terminal COOH was replaced with triazolyl. Three series of novel dual RT/IN inhibitors, including sulfide, sulfoxide, and sulfone analogs, were rationally designed by incorporating a DKA motif into the pyridin-2-one scaffold. The molecules consisting of ester or acetyl groups on the C3 position were found to have an improved binding affinity. The biological data of the compounds **89** and **90** showed that the sulfide series was the most effective dual-acting RT/IN inhibitors but the latter had more inhibitory action against IN, with an inhibition rate of 49.5% [99]. Researchers synthesized additional compounds in this series (compound **91**) as RT-associated RNase H/IN dual inhibitors due to high anti-HIV activity displayed by the quinolinonyl diketo acid derivatives consisting of a basic functional group like 1-pyrrolidinyl at position 7 (RNase H IC_50_ = 28 nM and IN IC_50_ = 5.1 µM). Compound **92** (RNase H IC_50_ = 3.3 µM and INST IC_50_ = 80 nM), **93** (RNase H IC_50_ = 6.8 µM and INST IC_50_ = 80 nM), and **94** (RNase H IC_50_ = 5.7 µM and INST IC_50_ = 50 nM) were found to be the most effective inhibitors for both enzymes among the recently synthesized quinolonyl diketo acid derivatives. Quinolinonyl N1-substitution with the 2,4-difluoro benzyl motif in compound **95** resulted in a high level of IN inhibitory action, according to the SAR analysis on quinolinonyl DKAs. However, compound **96** consisting of 2,6-dichloro benzyl was more effective in inhibiting RNase H [100,101].

1-[(2-hydroxyethoxy)methyl]-6-(thiophenyl)thymine (HEPT) was the first NNRTI candidate for clinical trials, and based on this compound several HEPT analogs have been synthesized as potential compounds for clinical studies. The X-ray crystallographic study suggests that the HEPT analogs exhibit comparable binding patterns and adopt a two-winged conformation in the HIV-1 NNRTI binding pocket [102]. The SAR analysis revealed that C5 alkyl had a considerable impact on the interactions of HEPT C6-benzyl with Tyr181. Additional van der Waals interactions by the 3,5-dimethyl group with the C6 benzene ring in the binding pocket were noticeably favorable to RT binding. The carbonyl group at the C6 position often resulted in decreased cytotoxicity and more powerful RT/IN dual actions [103]. Tang et al. presented dual HIV-1 RT/IN inhibitors, an extended series of N3- hydroxylated pyrimidine-2,4-diones with a benzoyl group at the C6 position of the pyrimidine ring with a low micromolar range activity. The HIV-1 RT and IN enzymes seemed to be effectively inhibited by compound **97**, which has a 3,5-dimethyl benzoyl motif replaced at position C6. The dimethyl group on the C6 benzene facilitates extra van der Waals contact with the RT binding site. On the other hand, it was discovered that the N1-benzene’s fluorination and the additional carbon in the N1 linker significantly reduced RT inhibition. The placement of fluorine or additional carbon to the N1 linker showed marginally favorable IN inhibition, which is an intriguing reversal of the tendency that was previously shown in IN inhibition [104].

Sun et al. created a novel series of pyrido[2,3-b]pyrazin-6 (5H)-one analogs (Figure 14) possessing C2 hydroxy group as dual-acting HIV-1 RNase H/IN inhibitors using the analog-based optimization of the divalent metal-chelating motif. The para-substituted benzene analog **98** exhibits inhibitory activity against RNase H and IN with IC_50_ values of 1.77 and 1.18 µM. The meta-substituted analogs were significantly more selective against RNase H than the para-substituted phenyl analogs. The exception was compound **99**, where it was discovered that substituting 4-cyanophenyl for 3-cyanophenyl decreased the dual RNase H/ IN inhibitory activity. This shows that well-structured hydrophobic and chelating pharmacophores yield an effective RNase H/IN dual inhibitor, with compound **98** as an example which has a balanced RNase H and IN micromolar inhibition [105].

Through the synthesis of various C7-substituted 2-hydroxyisoquinoline analogs, Billamboz et al. in 2008 discovered two novel hits **100** and **101** (Figure 14). These compounds showed promising selective IN inhibitory activity with submicromolar IC_50_ values (0.09 µM for **102** and 0.13 µM for **103**), as well as a moderate effect against RNase H at micromolar concentration [106]. Several coumarin-based analogs with a dual method of inhibiting RNase H/IN were discovered in research by Esposito et al., and compound **104** was found to have a potent antiretroviral activity with IC_50_ values of 6.25 and 6.45 µM for RNase H and IN, respectively [107].

In a study by Wang et al., the C5 methyl-sulfonamide group of the FDA-approved RT inhibitor DLV was swapped with a DKA motif to create novel RT/IN group dual inhibitors exhibiting sub- or low-micromolar action against RT and IN in enzymatic tests. Compound **105**, with IC_50_ values of 1.1 and 4.7 µM against RT and IN, respectively, showed balanced binding affinity against RT and IN. This class of compounds appears to adopt a spatial configuration that is advantageous for both RT and IN binding. According to SAR analysis, the hybrid ligands with the DKA on the C5 position, such as compound **106** (RT IC_50_ = 5.9 nM and IN IC_50_ = 12 µM) and **107** (RT IC_50_ = 120 nM and IN IC_50_ = 3.9 µM), adopted a pseudo-linear conformation similar to DLV, which is advantageous for both IN and RT binding. Additionally, it was identified that replacing the C3 position of the indole with bromine results in decreasing anti-RT and higher anti-IN activity, minimizing the discrepancy between the two activities [108,109].

Wang et al. used the presence of HEPT N1 benzyl near to the Pro236 at the NNRTI binding pocket to create a new generation of DKAs-HEPT analogs as RNase H/IN dual inhibitors. They coupled the DKA moiety of IN inhibitor with a TNK 651 generated from HEPT at N1 benzyl terminus. This combination had no negative impact on the intrinsic inhibitory action of any combined moiety. Based on the results, it was concluded that increasing the length of the N1 linker or replacing N1 with a different benzyl side chain interferes with the IN binding, indicating that only the C6 benzyl group satisfies the hydrophobic pharmacophore requirement for IN inhibition. Additionally, compounds **108** and **109** show that the linker between DKA and HEPT rings should be 2–3 atoms long in order to achieve the best RT/IN dual inhibition among HEPT derivatives [110].

The potential dual RT/IN inhibitors **110**–**113** (Figure 14) were created by Wang and Venice by incorporating the quinolone carboxylic acid at the N1 terminus of HEPT lead molecules. Of these compounds, compounds **110** (RT IC_50_ = 0.19 μM, IN IC_50_ = 35 μM) and **113** (RT IC_50_ = 3.7 μM, IN IC_50_ = 19 μM) showed the greatest RT and IN inhibitory activities [111].

### 2.14. Traditional Monotherapy

The NNRTIs are non-competitive, lipophilic, and highly bound to plasma protein with an affinity towards the allosteric binding pocket [112,113,114,115]. The traditional NNRTIs possess excellent EC_50_ values (NVP, 63 nM; ETR, 1.4–4.8 nM), good bioavailability (NVP >90%, DLV 85%, EFV 40–45%), and very good pharmacokinetic profiles. For example, the maximum plasma concentration (C_max_) for NVP is 2 μg/mL, EFV is 4.1 μg/mL, and for ETR and RPV it is 0.40 μg/mL and 0.15 μg/mL, respectively; T_max_ (time to reach maximum plasma concentration) for NVP, ETR, and RPV is 4 h, EFV is 5 h, and DLV is 1.17 h; and apparent volume of distribution (V_d_) ranges between 98–99.9%. The common side effects observed with NNRTIs are skin rash, headache, and nausea [116,117,118,119,120]. Single point mutations in the RT reduce the inhibitory potency of traditional drugs due to long-term usage. The point mutations such as L100I, K101P, K103N/S, V106A/M, V108I, Y181C/I, Y188C/L/H, G190A, and M230L are commonly observed with the RT enzyme as responsible for drug resistance [121,122].

The recently approved NNRTI DOR exhibits substantial inhibition against WT and mutant HIV-1. The common observed mutations, K103N, Y181C, and K103N/Y181C, are 95% inhibited with DOR concentrations of 43, 27, and 55 nmol/L (18.3, 11.5, and 23.4 ng/mL), respectively. Furthermore, there is an efficient inhibition of single mutant strains including A98G, E138A/G/K/Q, G190A, K101E/P, K103S, L100I, P236L, V106M, V108I, G190A V197D, V90I, Y181V, and Y188H/C. These mutant strains are not inhibited by the traditional drugs used for the treatment in the DRIVE-FORWARD, DRIVE-AHEAD, and DRIVE-SHIFT clinical studies. DOR has exhibited sustained antiviral efficacy in comparison to darunavir/ritonavir and EFV-based regimens. DOR possesses an elimination half-life of around 15 h, time to reach maximum plasma concentrations of 1–4 h, and a steady-state concentration time of 7 days. DOR showed very low resistant mutations (1.4%) in treatment-naïve patients (n = 9764), and the most prevalent mutations were V108I, Y188L, H221Y, and Y318F. DOR has an excellent safety profile as compared to the traditional drug EFV [123,124,125].

In 2017, the Russian Ministry of Health (MoH) approved an investigational drug Elsulfavirine (ESV or VM-1500A), developed by a San Diego-based biotech company, Viriom. The drug has an EC_50_ of 13.8 nM against a broad range of HIV-1 clinical isolates. The ESV has a longer half-life of ~9 days, making once-weekly treatment feasible. The prolonged activity of ESV is achieved by binding with RBC carbonic anhydrase and accumulation in RBCs causing a slow release of the drug, allowing for the development of long-acting oral and parenteral formulation. ESV (20 mg oral once-daily) in combination with tenofovir disoproxil fumarate (TDF)/emtricitabine (FTC) can be compared to EFV combination therapy in treatment-naïve HIV patients in a phase-2b clinical trial. A total of 81% of ESV patients show viral inhibition as compared to 74% of the patients treated with EFV. Moreover, the drug-induced adverse effects were minor in the ESV group (36.7%) as compared to the EFV group (77.6%). The maximum plasma concentration of ESV was 148 ± 8 ng/mL after 6.3 h [122,126].

### 2.15. Combination Regimen

Single-drug regimens with NNRTIs as a monotherapy are no longer prescribed due to the development of resistance. Instead, combination with other antiretroviral drugs [6] to overcome drug resistance is preferred over monotherapy. Combination therapy includes more than one class of antiretroviral agents targeting two or more enzymes in the life cycle of the virus [127,128]. EFV, RPV, and DOR are preferred for a combination regimen along with NRTI or INSTI. NNRTIs remain as part of the combination regimen owing to their unique mechanism of action, high specificity, and low toxicity. Moreover, the newer NNRTIs are highly active against mutant strains of HIV-1 compared to more traditional NNRTIs. The recommended HIV therapy suggested by the current national and international guidelines are three-drug regimens (3DRs) and two-drug regimens (2DRs) of NNRTI with other antiretroviral agents.

The fixed-dose combination (FDC) of cART consists of an NNRTI in combination with two approved RT inhibitors. The FDC reduces the dosing, ease of administration, and improves patient compliance, thereby improving the effectiveness and resistance profile. Seven FDCs have been approved to date, including Atripla (EFV, FTC, TDF) approved in 2006, Complera and Odefsey (FTC, RPV, TDF fumarate), approved in 2011 and 2016, respectively, and Symfi and Symfi Lo (EFV, lamivudine, TDF) approved in 2018. Since the cART regimens are prescribed for a longer duration, it can lead to associated problems such as drug–drug interactions, toxicity, and high cost [129,130,131,132,133,134]. The recently approved NNRTI, DOR, possesses fewer adverse effects and drug interactions, and was approved as a FDC along with FTC and TDF in 2019 under the brand name Delstrigo. NNRTIs like DOR, EFV, and RPV-based regimens, in combination with NRTIs or NtRTIs, remain the prime choice as a first-line treatment where INSTI-based regimens cannot be applied [135,136,137]. In cART, compared to the triple regimen, the dual regimen of dolutegravir (DTG) and RPV in combination (Juluca) without protease inhibitors shows a high rate of 90% virological suppression at 48 weeks, avoiding metabolic adverse effects in bone, lipid, kidney, and drug interactions, making it a better choice for aging HIV-infected patients [138]. Clinical studies have revealed that a combination of DTG and DOR when given to healthy subjects displayed significant pharmacokinetic and safety profiles with no interactions. This promotes DTG and DOR as a FDC for HIV-1 therapy. It would be interesting to clinically test the combination of INSTI/NNRTI. Novel INSTI/NNRTI combinations are among possible future antiretroviral drug approvals [137].

### 2.16. Formulation

HIV patients are susceptible to high resistance problems due to poor adherence to life-long oral therapy. Oral therapy suffers from drawbacks such as continuous medication, poor drug adherence, and missed doses, which are serious concerns leading to viral relapse. As a result, long-acting dosage forms need to be developed for improving therapeutic outcomes and patient compliance. The suitable long-acting dosage forms for HIV treatments are oral, parenteral, and implant. These dosage forms help to achieve once weekly for oral, once monthly for parenteral, or once 6-monthly dosing, respectively [139].

The long-acting (LA) injectable form for antiretroviral candidates is being investigated as an alternative approach to traditional oral daily therapy. The slow drug release of LA injections can maintain a constant drug concentration in plasma through a single dose [140,141]. Non-injectables such as implants, patches, and vaginal rings can also deliver antiretrovirals for long durations. The ability of these LA dosage forms to release the drug in a prolonged manner determines the frequency of administration. However, the mode of release from the LA dosage form varies from depot injection (e.g., RPV as suspension) to polymer erosion-controlled mechanisms (e.g., Islatravir as implants), and mucoadhesion (Dapivirine) [142]. In addition, they also act as depots and aid in delivering ARV to organs with fewer accessibility sites like the brain and thymus. Complications like regular monitoring, more frequent clinic visits, high drug loading, painful injections still exist with LA formulation. Nevertheless, many NNRTI’s as LA formulations are still in the developmental stage with Cabenuva being approved recently and can be expected in the next decade (Table 1).

Drug adherence to multiple drugs is essential to prevent relapse, and hence the dosage regimen by extending dosing intervals by LA formulation may lead to a successful therapeutical approach. Recently, ViiV Healthcare and Janssen Pharmaceuticals (Janssen) developed a dosing kit containing separate injectable suspensions of Cabotegravir and RPV for intramuscular (IM) use (Table 2). The patient’s tolerance was assessed using once-daily oral tablet Cabotegravir (30 mg) and oral RPV (25 mg) for 30 days. This was followed by an injectable dose (3 mL dosing kit with 600 mg Cabotegravir and 900 mg RPV) on gluteal muscles and a maintenance dose every month (2 mL dosing kit with 400 mg Cabotegravir and 600 mg RPV). Both the antiviral agents exhibited a flip-flop absorption mechanism leading to slow absorption from the gluteal muscle into systemic circulation, eventually attaining the required sustained plasma concentrations. The once-daily formulation and drug plasma ratios were superior to the PK trials of LA, and both medications accumulated 2- to 2.3-fold between the troughs at the eighth and 48th week, with steady-state concentrations. During trials, severe post-injection reactions including dyspnea, agitation, abdominal cramping, flushing, sweating, oral numbness, and changes in blood pressure, were observed in <0.5% of subjects due to improper intravenous administration.

Clinical trial data for the Phase III Antiretroviral Therapy as Long-Acting Suppression (ATLAS) and First Long-Acting Injectable Regimen (FLAIR) trials confirmed that the combined LA injection was as effective as standard once-daily three-drug therapy in HIV suppression (two NRTIs in addition to INSTI, NNRTI, or PI). After 96 weeks, viral suppression was maintained at 87% and a considerable percentage of patients exhibited injection site reactions (mild 84%, moderate 15%). Nevertheless, the dosage form was discontinued in <1% of the patients due to injection site-related adverse events [145,146]. The LA kit was approved by the FDA on January 21, 2021, as a co-packaged complete regimen for HIV-1 infected adults that is injected once a month by intramuscular route. Cabenuva is available as a monthly co-package with two injectable medicines—ViiV Healthcare’s Cabotegravir and Janssen’s RPV—as an option to replace the current antiretroviral regimen in people who are virologically suppressed (HIV-1 RNA less than 50 copies per milliliter) on a stable regimen, have no history of treatment failure, and have no known or suspected resistance. To check the tolerance of each medicine, oral dosing of Cabotegravir and RPV should be given for around one month before starting Cabenuva treatment. Individuals must attain an undetectable HIV RNA level before changing to the long-action, injectable combination therapy, that is offered every 4 or 8 weeks in clinical studies of intramuscular Cabotegravir + RPV. The authorized FDA dosing is every 4 weeks. People can take oral forms of Cabotegravir and RPV in order to confirm tolerability before transitioning to the long-acting formulations. Careful selection of patients who adhere to injectable therapy more closely and maintain the systemic concentration of medications at a standard level should be taken into consideration in order to reduce the likelihood of Cabotegravir/RPV resistance. Future research to create additional injectable LA ART to lessen the frequency of medication intake can be sparked by Cabotegravir/RPV [147].

Furthermore, subcutaneous implants as LA formulations of NNRTIs like MK-8591 are being investigated in the clinical scenario and their results are expected in this decade. The implant used ethylene vinyl acetate as polymer, a non-biodegradable removable polymer for controlling the systemic drug release from the implant [148]. Preclinical reports of the drug-eluting implant were positive and it was efficient in maintaining plasma levels of MK-8591 for 6 months in murine and primate models [149]. In human trials, 62 mg of the Islatavir-loaded implant was able to maintain the required plasma concentration for 12 months (pharmacokinetic threshold > 0.05 pmol/10^6^ cells) without significant adverse events [150].

Surve et al. prepared long-acting Efavirenz--enfuvirtide co-loaded polymer lipid nanoparticles capable of releasing the medicament in murine models. The subcutaneous dosage form elucidated a favorable distribution pattern where the liver and spleen act as a drug reservoir. The dosage form was able to deliver the antiviral payload to infected sites like lymph nodes and the reproductive tract for 5 days. Interestingly, the nanoparticles also permeated the blood-brain barrier and the results were promising to employ nano-formulations for LA delivery in human use [144]. For the delivery of a synergistic two-drug ARV combination consisting of a pre-clinical NNRTI, Compound **55**, and NRTI, 4′-ethynyl-2-fluoro-2′-deoxyadenosine (EFdA), Beelor et al. developed two LA-ART interventions, one an injectable nano-formulation and the other based on pharmacokinetics and antiviral activity in a hu-mouse model of HIV infection. Furthermore, the Compound **55**/EFdA combination was separately manufactured as LA PLGA-based nano-formulations (LA-NP) and poly(PDL-co-DO) implants (LA-implant). The LA-NP and LA-implant formulations displayed consistent plasma levels of both medications throughout the duration of the research. At day 49 in hu-mice with nano-formulations, Compound **55** and EFdA (Figure 7) in sera were 2.0 µg/mL (4 µM) and 4 µg/mL (13 µM), respectively, and 2 µg/mL (6.8 µM) and 7.5 µg/mL (25 µM), respectively, at day 56 in hu-mice with nanoformulations and implants. The PK data analysis showed that the Compound **55** NP had a 2.5-fold lower AUC_0-last_ than the Compound **55** implant due to the roughly twice greater dosage attained with the Compound **55** implant. Both the Compound **55** formulations had the same clearance. The slower metabolism of Compound **55**, as shown by the increased volume of distribution and low CL rate, enables sustained levels for viral control, as demonstrated in earlier research using free Compound **55** [151].

## 3. Comparative Assessment of the Strategies in the Development of Novel NNRTIs

Over the last three decades, significant advancements have been made in the development of NNRTIs. At this time, six NNRTIs have secured FDA approval for the treatment of HIV-1, including nevirapine (NVP, 1996), delavirdine (DLV, 1997), efavirenz (EFV, 1998), etravirine (ETR, 2008), rilpivirine (RPV, 2011), and doravirine (DOR, 2018) (Figure 15). Despite the efficacy of NNRTIs in treating and preventing HIV-1, their effectiveness has significantly declined due to mutations occurring in or around the NNIBP. An ideal anti-HIV drug should meet specific criteria: (i) enhanced activity against both wild-type and drug-resistant viruses; (ii) favorable oral bioavailability and metabolic stability; (iii) minimal side effects and a favorable safety profile; (iv) absence of drug–drug interactions; and (v) ease of synthesis and formulation. Nonetheless, novel pharmacological approaches, innovative formulation strategies, and medicinal chemistry tactics have emerged to address drug resistance and issues related to solubility. These approaches also promote the identification and optimization of potential antiviral therapeutic agents.

Integrated modern medicinal chemistry strategies such as molecular/fragment hybridization, bioisosterism, scaffold and fragment hopping, conformational restriction, prodrug development, ligand lipophilic efficiency, high-throughput screening (HTS), covalent binding, targeting highly hydrophobic channels, dual-site targeting, and the use of multi-target-directed ligands have resulted in the creation of several highly potent analogs. Notably, NVP, the first licensed NNRTI drug, primarily employed two classical medicinal chemistry approaches, molecular hybridization and bioisosterism, during its lead optimization process. DLV mesylate, also known as PNU-90152, was initially discovered in 1992 by Upjohn scientists using a high-throughput screening (HTS) strategy from a chemical library containing 1500 structurally diverse compounds. This compound exhibited remarkable structural similarities to bis(heteroaryl)piperazine (BHAP). Additionally, the benzothiadiazine compound NSC287474 served as a prototype for the development of EFV and displayed strong efficacy against various HIV-1 strains. The lead compound L-608,788 was synthesized through bioisosteric replacement during EFV’s development.

The TMC120 molecule represents a novel prototype diarylpyrimidine (DAPY) analogue of ETR and RPV, belonging to the new class of diaryltriazine (DATA, R106168), where the central triazine ring replaced the bioisostere, pyrimidone. The combined use of molecular/fragment hybridization and bioisosterism strategies presents a rational approach to NNRTI drug development, aiding in the battle against escalating drug resistance. The first-generation NNRTIs, including NVP, DLV, and EFV, demonstrated reasonable efficacy compared to earlier treatments. However, setbacks, such as DLV’s inefficacy when used alone, necessitated the use of combination therapies. All these compounds possess a low genetic barrier to resistance, making it possible for effective drug resistance to arise from a single mutation in the RT sequence.

To address RT mutations more effectively, second-generation NNRTIs, including ETR, RPV, and DOR, were developed. These drugs outperformed their first-generation counterparts in terms of inhibitory activity, especially against HIV mutations. Nevertheless, the development of NNRTIs continued due to resistance to these second-generation drugs, RT mutations, dose-limiting toxicity, and undesirable pharmacokinetic characteristics. Current medicinal chemistry approaches can facilitate the discovery of new NNRTIs primarily focused on DAPY structural modifications, DAPY mimic scaffolds, peripheral substituents, and a limited number of other derivatives as potential HIV-1 RT inhibitors.

Various techniques have been employed in the development of promising novel NNRTIs, including molecular hybridization, structure-based bioisosterism, scaffold hopping, conformational restriction, fragment hopping, targeting highly hydrophobic channels, and hydrophilic solvent-exposed area drugs. These diverse NNRTI analogs (Appendix A, also available under zenedo.org, accessed on 20 September 2023), including diarylbenzopyrimidine (DABP), pyridine-dimethyl-phenyl-DAPY, piperidine-substituted thiophene[2,3-d]/[3,2-d]pyrimidine, hydroxyl-substituted biphenyl-diarylpyrimidines, CH(CN)-DAPYs, dihydrothiopyrano[4,3-d]pyrimidine, thiophene-biphenyl-DAPY, racemic diarylpyrimidine CH(OH)-DAPY, methylsulfonyl sulfamide-substituted piperazine/piperidine DAPY, dihydrofuro[3,4-d]pyrimidine, naphthyl catechol phenyl ether, pyridyl-bearing fused bicyclic, indolylarylsulfones (IASs)-bearing phenylboronic acid, sulfur-containing diarylbenzopyrimidine, difluoro-biphenyl-diarylpyrimidine, phenylaminopyridine, and 1,2,3-triazole-derived diarylpyrimidine analogues, have exhibited high resistance profiles, low toxicity, enhanced efficacy, improved specificity, increased solubility profiles with good oral bioavailability, reduced cytochrome liabilities, and minimal human ether-a-go-go gene (hERG) inhibition, making them promising candidates for HIV clinical therapy. Their structural adaptability and the establishment of numerous hydrophobic connections, which in turn create hydrogen-bond interactions within the NNRTI binding pocket, have also demonstrated a strong binding affinity. Consequently, the prospective approaches provide valuable therapeutic interventions for addressing various HIV infections.

Over the past two decades, there has been a substantial upsurge in clinical experience with combination therapy strategies for managing HIV infection. These strategies have exhibited comparable efficacy, safety, and convenience when compared to monotherapy, both in individuals new to antiretroviral therapy (ART) and those with prior treatment experience who have achieved virological suppression.

Three-drug regimens (3DRs), based on either protease inhibitors (PIs) or non-nucleoside reverse transcriptase inhibitors (NNRTIs) combined with two nucleoside reverse transcriptase inhibitors (NRTIs), have served as the standard of care for HIV therapy since 1996. Despite the presence of various side effects and ART-related toxicities, the success of first-generation 3DRs has substantially increased the life expectancy of individuals living with HIV.

The development of innovative two-drug regimens (2DRs) commenced around the mid-2000s, with the objective of reducing adverse effects, minimizing drug interactions, and enhancing patient adherence. Current HIV clinical guidelines advocate for the use of 2DRs as the primary treatment option in several clinical scenario. This recommendation is supported by numerous clinical trials comparing the efficacy of 2DRs and 3DRs in both treatment-experienced and treatment-naive patients, demonstrating that 2DRs are on par with 3DRs in terms of effectiveness.

Persistent research efforts have led to the formulation of a novel drug strategy for Cabenuva™ which has recently been approved by the FDA. This formulation includes long-acting (LA) cabotegravir combined with LA RPV nanocrystals, developed by ViiV Healthcare. It is administered once monthly via intramuscular injection, promoting treatment compliance. Additionally, Cabenuva™ offers new pre- and post-exposure prophylaxis, which may play a crucial role in curtailing the spread of the disease, particularly in populations disproportionately affected by the pandemic.

In contrast to conventional drug regimens, the LA nanoformulation prioritizes increased stability, lymphatic transmission, and gradual release into the systemic circulation. Several medicinal chemistry techniques, combination therapy protocols, and formulation strategies have significantly enhanced the efficiency of various stages in the development of NNRTI-based anti-HIV drugs.

## 4. Conclusions

Reverse transcriptase is one of the well-established targets for HIV drug discovery. Several pharmaceutical strategies (medicinal chemistry, pharmacology, and formulation) were successfully applied to improve the HIV-1 inhibition potency in cell-based and enzymatic assays by effectively blocking the polymerization of WT and mutant types, to resolve the poor ADME properties and to reduce the toxic profiles. There is an increasing level of drug resistance and adverse side effects during clinical treatment using NNRTIs for different types of HIV-infected patients. Due to this, there is a need for combination strategies involving NNRTIs with other antiretroviral targets for efficient treatment of HIV-1. Despite the success of the combination regimens, there are certain limitations for life-long treatment which include increase in cost, drug resistance, life-long adherence to therapy, and unknown adverse effects. Recent clinical studies focus on the dual combination and long-acting formulation of NNRTIs along with integrase inhibitors. These approaches improve patient adherence, and show higher virological response, better drug resistance profiles, and fewer adverse events, especially for difficult-to-treat patients. NNRTIs are significant for HIV-1 treatment and prevention in resource-limited countries. A deep understanding of the PK/PD properties of NNRTIs provides a greater advantage for new drug development and optimization of multi-drug regimens. A simple multidrug approach may not be as effective as a single molecule with several functions. The ability to stop viral replication in two or more phases makes the creation of these multifunctional ligands as antiretroviral drugs valuable. The progression of NNRTIs and other anti-HIV drugs will transform HIV infections into a controllable disease and combat them efficiently.

## 5. Perspective

The reverse transcriptase enzyme is an attractive target for the treatment of AIDS because of its elementary molecular mechanism in virus replication and pathogenesis of HIV-1 virus-infected patients. For the last two decades, NNRTIs have been developing into a keystone for the cure of HIV infection. This diverse chemical class of NNRTI drugs blocks the reverse transcription non-competitively at the allosteric site of RT by disruption of the enzyme’s catalytic site. The NNRTIs are classified into two generations, depending on their discovery and resistance profile. The first-generation NNRTIs, NVP, and DLV, possess rigid structures while the second-generation EFV, ETR, RPV, and DOR are more flexible, causing the right adaptation inside the non-nucleoside RT binding pocket, and offer a greater barrier to the development of resistance as compared to earlier generations. The second generation has unique anti-HIV activity, high specificity, and low toxicity, but their therapeutic efficacy is decreased by single- or double-point mutations occurring in or near the NNIBP after prolonged treatment.

A variety of medicinal chemistry strategies along with computational approaches including molecular hybridization, bioisosterism principles, scaffold hopping, fragment hopping, conformational restriction, prodrug approach, high-throughput screening (HTS), covalent-binding, targeting highly hydrophobic channels, targeting dual sites, application of multi-target-directed ligand fragment-based screening, structure-based screening, and ligand lipophilic efficiency have been applied successfully to the discovery of NNRTIs for achieving higher potency against a wide range of resistant strains. All the strategies mainly focus on modulating the structural features such as linker/rotatable bonds, varying chiral centers, and hydrophobic and polar spaces for maintaining favorable interactions at the allosteric site of RT including the various mutant strains and enhance ADME properties (water solubility, metabolic stability, and oral bioavailability).

Combination regimens of NNRTIs are highly recommended with other classes of anti-HIV drugs. The introduction of NNRTIs in cART plays an essential role in the suppression of HIV-1 RNA, acquires a synergism effect, reduces individual drug toxic side effects, conserves CD4 immune function, minimizes drug resistance, diminishes HIV-1 associated morbidity, impedes HIV spread, has pharmacokinetic benefit, and improves health. At present, one of the three drugs from NNRTI (EFV or RPV or DOR) combined with other antiretroviral drugs as double- or triple-regimen (Atripla, Symfi, Symfi Lo, Complera, Odefsey, Delstrigo, and Juluca) are capable of reducing the HIV-1 infection from a deadly infectious disease to a controllable chronic disease. The dual regimen such as DTG with RPV combination (Juluca) without protease inhibitors is particularly promising in aging HIV-infected patients. Although cART is superior to monotherapy, there are some limitations, such as cost of long-term treatment, drug resistance, lifetime follow-up, and uncertain consequences over long-term application. The recently approved drug DOR shows good antiviral potency against the single mutant viruses, including A98G, E138A/G/K/Q, G190A, K101E/P, K103N/S, L100I, P236L, V106M, V108I, G190A V197D, V90I, Y181C/V, and Y188H/C, as well as double mutant K103N/Y181C strains. This clearly demonstrates that DOR is a highly potential NNRTI in combination with other classes of anti-HIV drugs; a successful example of this approach is Delstrigo which shows promising results in clinical trials. Moreover, DOR is under investigation in various combinations such as DOR/Raltegravir, DOR/Lamivudine (3TC)/TDF, and Islatravir (MK-8591)/DOR/Lamivudine (3TC) for the treatment of HIV-1 infected children and adolescents, and is also being explored in patients with virological failure due to EFV-based antiretroviral therapy. The combination could also help to control viral replication, minimizing the occurrence of long-term clinical and metabolic complications, and the risk of drug–drug interactions.

The first long-acting injectable antiretroviral therapies, Rekambys (RPV) and Vocabria injection (Cabotegravir), are recommended for approval in the treatment of patients with HIV-1 infection. The ATLAS clinical trial studies concluded that long-acting, injectable Cabotegravir/RPV, dosed every 8 weeks, is an effective and well-tolerated formulative approach to maintain viral suppression. The Cabotegravir/RPV combination therapy persists in people even after the last injection of the drug, raising the risk of drug resistance and potentially causing toxicity and serious side effects. Additional investigational medications are in the pipeline for possible long-acting injectable therapy which includes oral and injectable (subcutaneous) formulations of the capsid inhibitor Lenacapavir, and a long-acting injectable formulation of VM1500A, a novel potent broad-spectrum NNRTI in early phase trials for HIV maintenance therapy. The success of long-acting injectables in clinical trials will provide a new tool for HIV therapeutic treatment. Furthermore, it will improve adherence, increase convenience, and help reduce the stigma associated with daily pill-taking for people with or at risk of HIV. These novel approaches help us realize the goals of initiatives for ending the HIV Epidemic (EHE) and achieving viral suppression and preventing new transmission. New formulations (topical microbicides, nano-formulations, and extended-release) and co-formulations of two or more antiretroviral drugs for improving HIV-infected patient compliance will be part of formulation research. It focuses on less frequent dose administration as well as increasing drug concentrations, completely eradicating HIV at the specific reservoir sites such as vaginal tissue, rectal tissue, and sites in the immune system.

Currently, the application of advanced biophysical techniques such as X-ray crystallography, NMR spectroscopy, and SPR-based biosensor analysis provides structural information on RT for WT and different mutant strains along with various scaffolds of NNRTIs. This structural information brings insight into the conformational change of amino acid residues in the binding pocket and their interaction with NNRTIs. The combined usage of medicinal chemistry strategies and structural information obtained from biophysical techniques helps to design novel NNRTIs with conformation restriction in the binding pocket, focusing on conserved residues and interaction with the amino acid backbone to overcome the mechanism of resistance. The recent scientific advancements in big data enable us to store, analyze, and utilize the information to apply artificial intelligence techniques including machine learning and deep learning to predict bioactivity, physicochemical property, and PK/PD information. These current advancements in artificial intelligence, biophysical techniques along with various pharmaceutical strategies (medicinal chemistry, pharmacology and formulation), and the collected data on HIV targets and inhibitors widens the opportunity for the effective design and development of NNRTIs with high potency against resistant strains and low toxicity profiles.

## Figures and Tables

**Figure 1 viruses-15-01992-f001:**
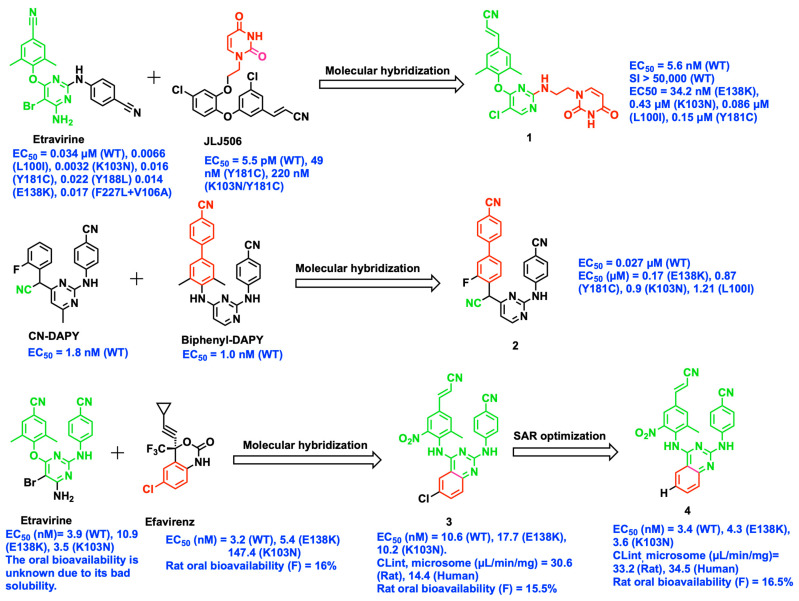
Molecular hybridization strategy. Optimization of uracil-substituted DAPY, biphenyl-DAPY with a cyanomethyl linker, diarylbenzopyrimidines.

**Figure 2 viruses-15-01992-f002:**
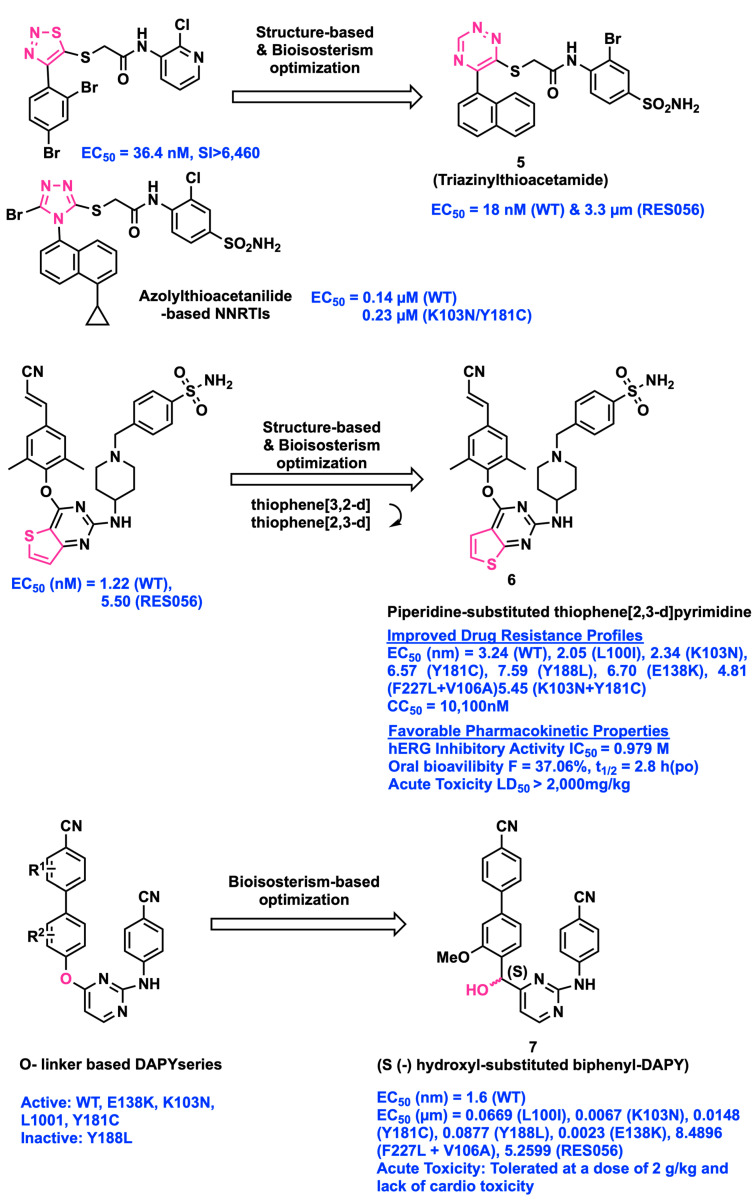
Bioisosterism strategy: Combination of bioisosterism with structure-based optimization of triazinylthioacetamide, piperidine-substituted thiophene[2,3-*d*]pyrimidine, and chiral hydroxyl-substituted biphenyl-diarylpyrimidine derivatives identified as potent HIV-1 NNRTIs.

**Figure 3 viruses-15-01992-f003:**
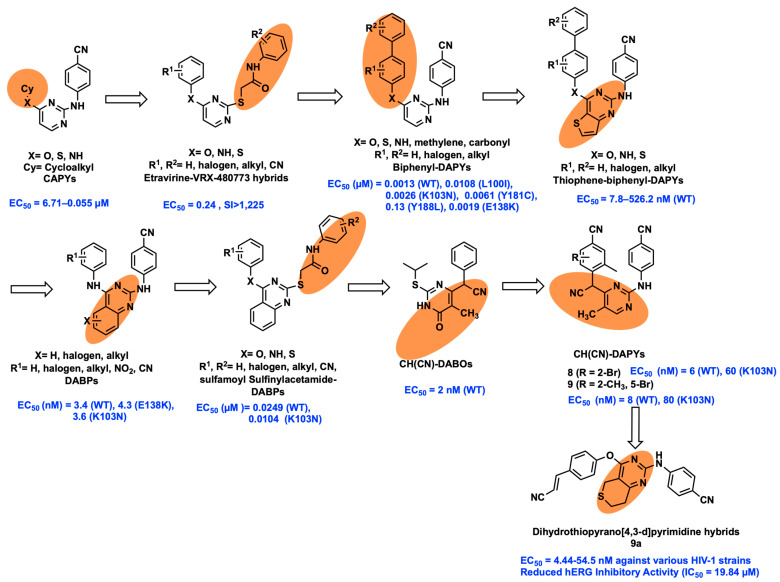
Scaffold hopping strategy. Evolution of diverse DAPY-like derivatives.

**Figure 4 viruses-15-01992-f004:**
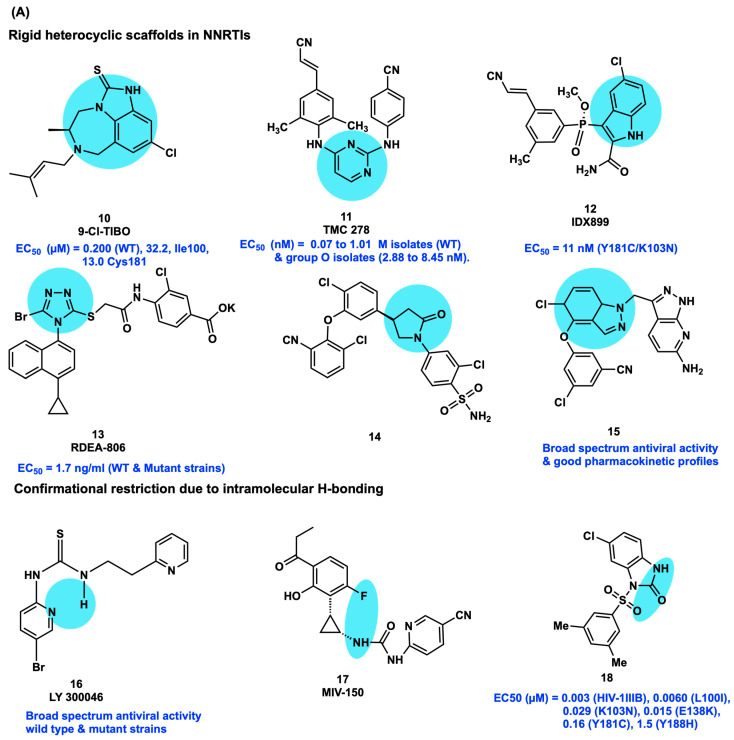
Conformational restriction strategy. (**A**) Design of rigid heterocyclic scaffolds and intramolecular H-bonding as restricted factors for maintaining the “butterfly-like” conformation. (**B**) Design of conformationally restricted scaffolds by using chiral cyclopropane rings as restricted factors for maintaining the “butterfly-like” conformation in the binding pocket of RT. (**C**) Design of conformationally restricted derivatives by using stereochemical asymmetric geometry and extending dihedral angle as restricted factors for maintaining the “butterfly-like” conformation in the allosteric site of RT.

**Figure 5 viruses-15-01992-f005:**
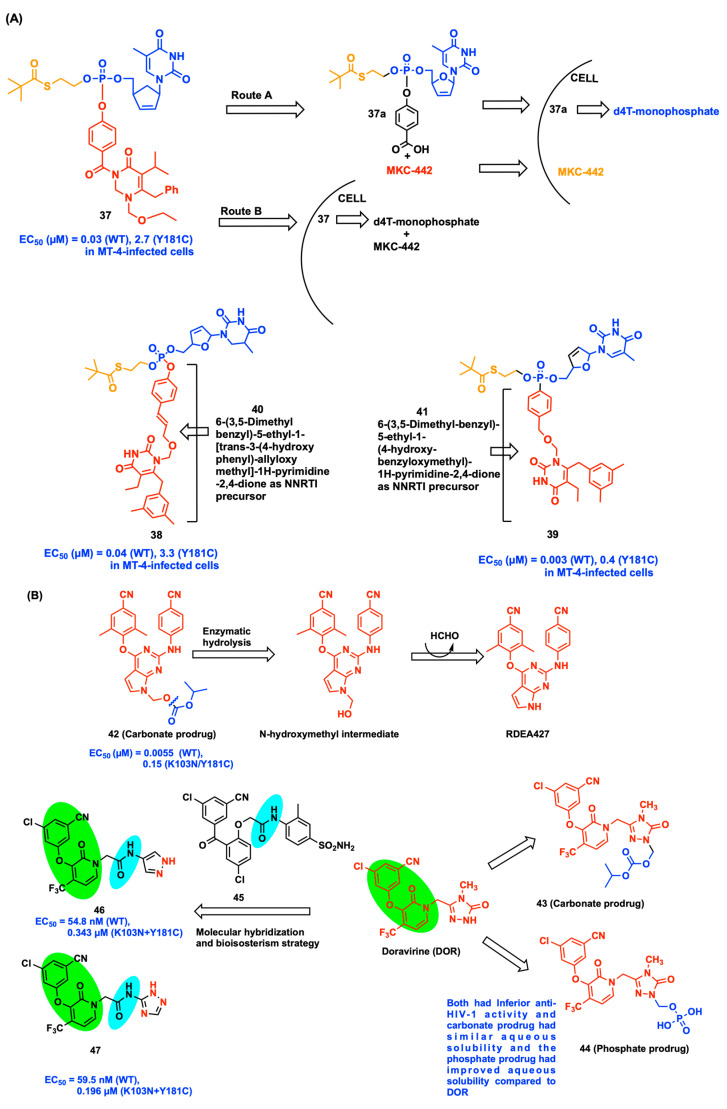
Prodrug strategy. (**A**) Development of double-prodrugs by coupling D4T (NRTI) with Emivirine (NNRTI). (**B**) Design and synthesis of acetamide-substituted DOR and its prodrugs as potent HIV-1 NNRTIs.

**Figure 6 viruses-15-01992-f006:**
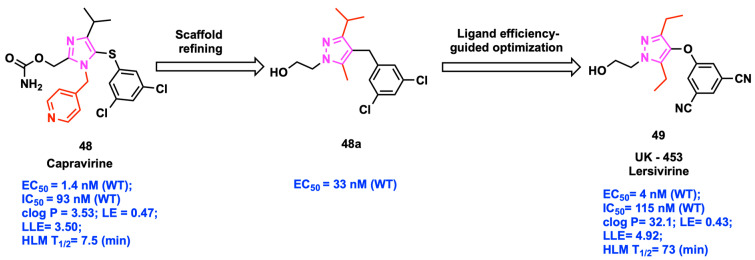
Ligand efficiency strategy. Optimization of clinical candidate molecule Lersivirine on the basis of Capravirine scaffold refinement.

**Figure 7 viruses-15-01992-f007:**
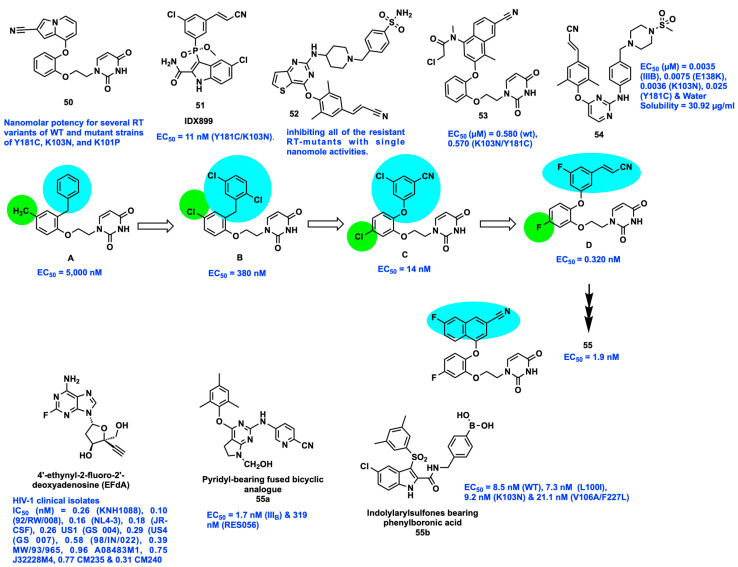
Structure-based optimization strategy. Design and development of novel aryl-ether scaffolds, pyridyl bearing fused bicyclic and indolylarylsulfones analogues by utilizing the crystal structures of RT.

**Figure 8 viruses-15-01992-f008:**
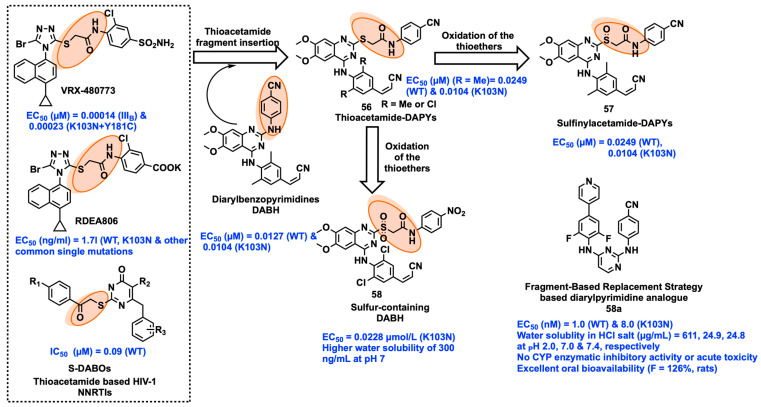
Fragment hopping strategy. Flowchart of identification and optimization of thioacetamide dibenzopyrimidines as HIV-1 NNRTIs via the privileged fragment-based reconstruction approach.

**Figure 9 viruses-15-01992-f009:**
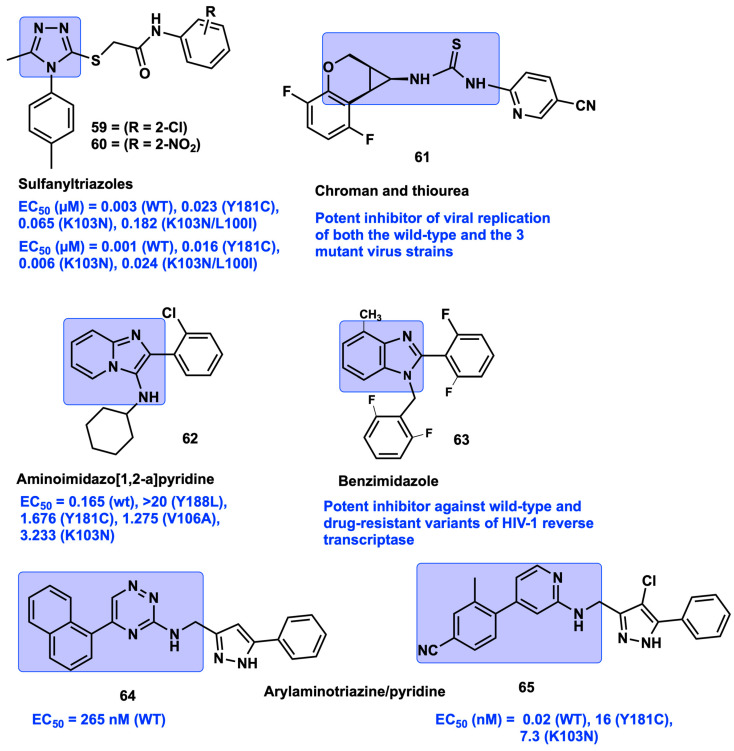
High-throughput screening strategy. Schematic representation of the high-throughput screening approach for discovery of potent diverse scaffold HIV-1 RT inhibitors.

**Figure 10 viruses-15-01992-f010:**
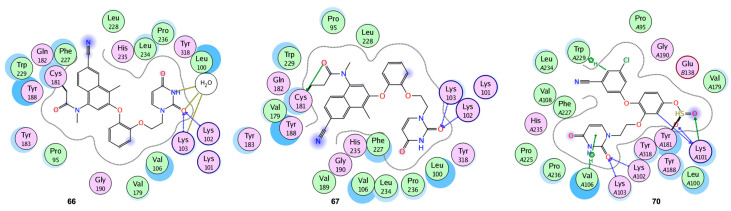
Compounds **66** (**left**) and **67** (**middle**) form a covalent bond inhibition with the thiol group of Cys181 in the HIV-1 Y181C RT (PDB codes: 5VQX and 5VQV). Covalent inhibition of compound **70** (**right**) to wild-type HIV-1 Reverse Transcriptase at Tyr181 residue using a fluorosulfate warhead (PDB 7KRD).

**Figure 11 viruses-15-01992-f011:**
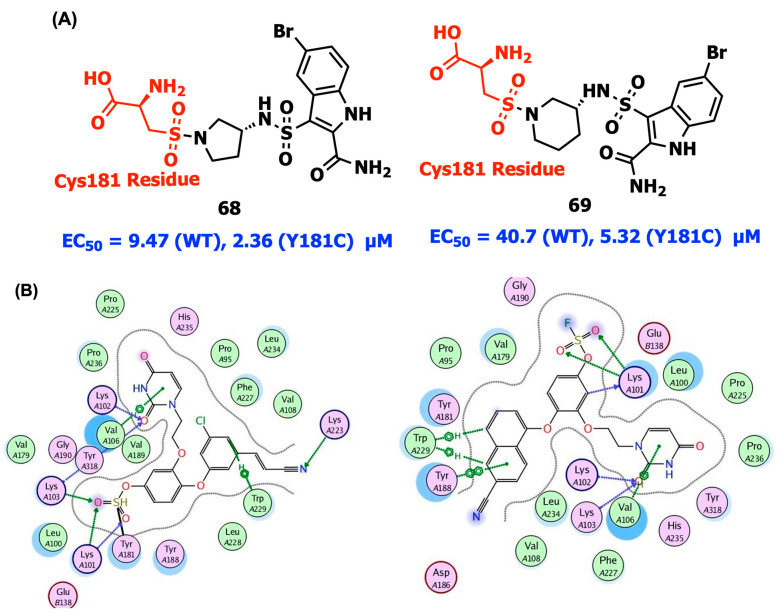
(**A**) Covalent Bonding Inhibition. Compounds 68 and 69 form covalent bonds with the Cys181. (**B**) The crystal structure for **71** (**Left**) and **72** (**Right**) is covalently bound to the hydroxyl oxygen atom of Tyr181 and side chain nitrogen atoms of Lys101 in the wild-type HIV-1 reverse transcriptase.

**Figure 12 viruses-15-01992-f012:**
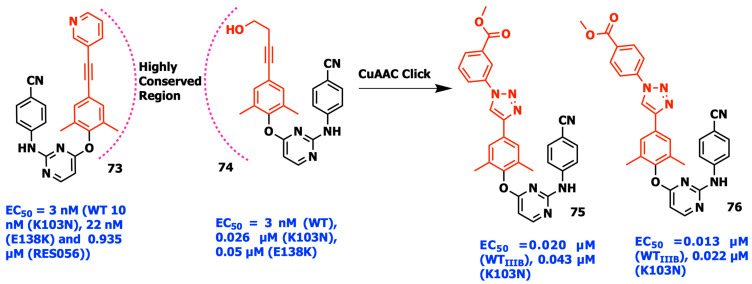
Targeting highly hydrophobic channels. Highly conserved regions of **73** and **74** which further developed into para-methylbenzoate analog **75** and the metamethylbenzoate analog **76** by CuAAC click reaction.

**Figure 13 viruses-15-01992-f013:**
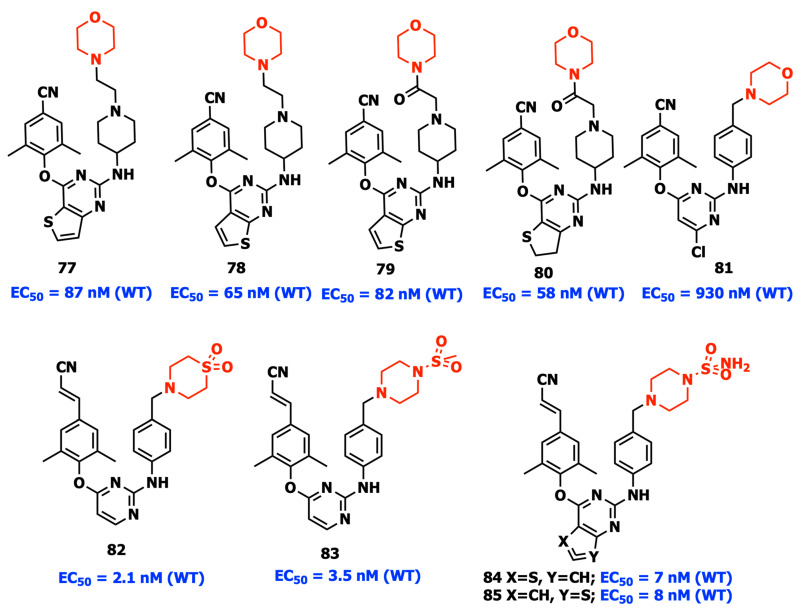
Targeting the hydrophilic solvent-exposed region. The potential binding solvent-exposed regions (red-colored region), which possibly accommodate structurally diverse moieties and form additional interactions, provide broad chemical space for substantial modifications.

**Figure 14 viruses-15-01992-f014:**
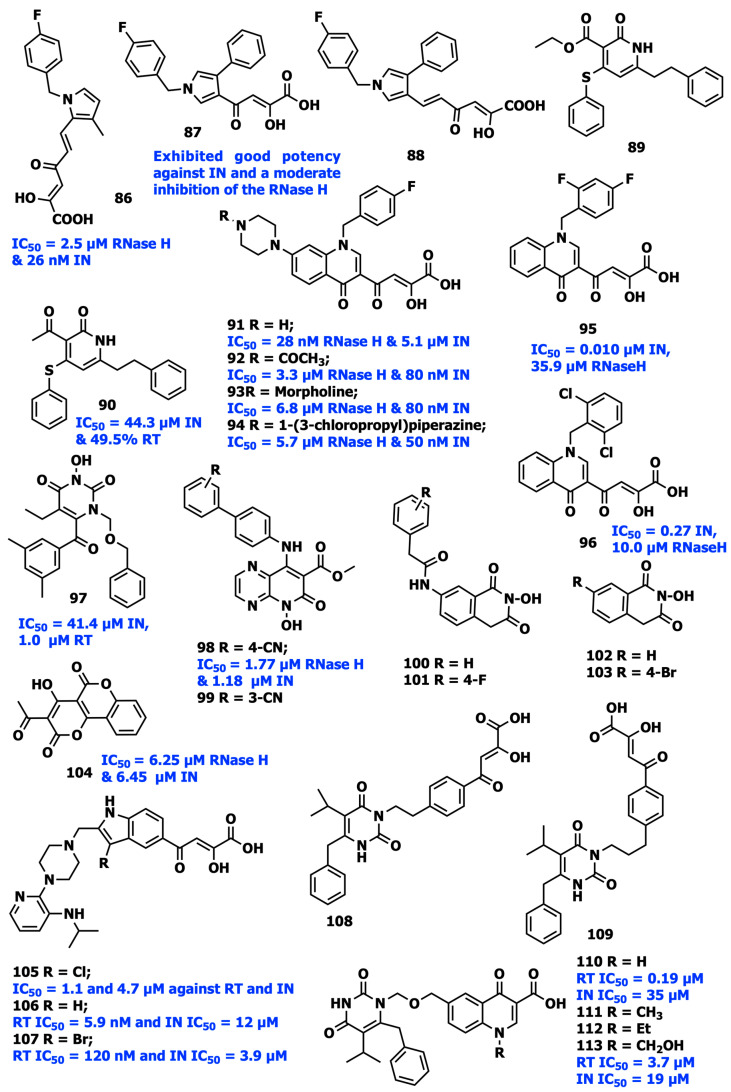
Targeting dual sites. Various inhibitors targeting dual enzymes, RNase and IN.

**Figure 15 viruses-15-01992-f015:**
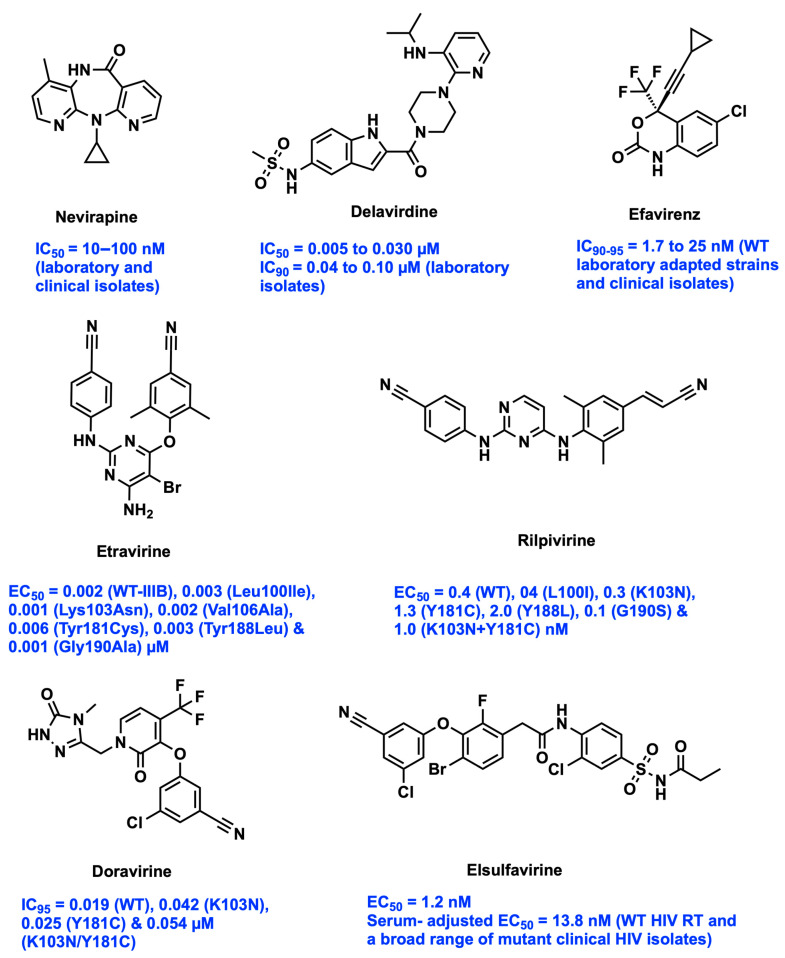
FDA approved NNRTIs. The approved drugs for NNRTI from 1996 to 2018.

**Table 1 viruses-15-01992-t001:** NNRTI-based long-acting approaches and their clinical status for the treatment and prevention of HIV.

Brand	Drug	Clinical Status	Administration
Cabenuva	Cabotegravir + Rilpivirine	Approved	Suspension Injection
Elpida	Elsulfavirine	Preclinical	Oral formulation [143]
-	Efavirenz (Polymer lipids with nanoparticles co-loaded with Enfuvirtide)	Preclinical	Subcutaneous [144]

**Table 2 viruses-15-01992-t002:** NNRTI- and integrase-based long-acting formulations and their pharmacokinetic data.

Brand	Edurant	Rekambys	Vocabria	-
Drug	Rilpivirine	Rilpivirine	Cabotegravir	Cabotegravir
**Route of** **Administration**	Oral	Injection	Oral	Injection
**Strength**	25mg	900mg ^i^; 600mg ^m^	30mg	600mg ^i^; 400mg ^m^
**T_max_**	4–5 h	3 to 4 days	3 h	7 days
**Blood-to-plasma ratio**	0.7	0.7	0.5	0.5
**t_1/2,_ mean**	45–50 h	13 to 28 weeks	41 h	5.6 to 11.5 weeks
**Subsequent-dose pharmacokinetic**				
**C_max_**	247 ng/mL	116 ng/mL	8.1 µg/mL	4.2 µg /mL
**AUC_tau_**	3300 ngh/mL	65603 (ngh/mL)	146 µg/mL	2461 µg /mL
**Time taken**	Day 7	44 weeks	Day 7	44 weeks

^i^ = initiation dose; ^m^ = maintenance dose; T_max_ = Time of Maximum concentration; t_1/2_ = half-life; AUC = Area under the curve; Time taken = Time taken to achieve steady state concentration C_max_ = maximum serum concentration; h = hours; ng/mL = nanogram/milliliter; ngh/mL = nanogram hour/milliliter; µg/mL = microgram/milliliter.

## Data Availability

No new data is created. The data compiled in this is available without any restrictions under the http://www.zenodo.org (https://doi.org/10.5281/zenodo.8349480), accessed on 20 September 2023.

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
