# Peer review of "Strategies in the Design and Development of Non-Nucleoside Reverse Transcriptase Inhibitors (NNRTIs)"

_viruses, 2023, doi:10.3390/v15101992_

Round 1

Reviewer 1 Report

Dear Authors.

The manuscript contains a thorough review of the strategies used in drug discovery in general and provides appropriate examples of their application to the design of NNRTIs.

I think that in its current form the manuscript contains a lot of chemical information that needs to be recomposed to improve the summary for each of the strategies considered and a comparison of these strategies in the development of (a) drugs in general and (b) NNRTIs. In particular, it would be great if the authors could include the number of drugs, and especially NNRTIs, whose activity has been confirmed in experimental studies for each of the strategies and discuss the results. An additional comparative study of clinical trials for the development of NNRTIs would be an additional feature of the manuscript.

The volume can be reduced by focusing on the critical evaluation of the strategies and keeping the details that allow the reader to see the relationship between the strategy and its critical evaluation provided by the authors.

Therefore, I suggest the following steps to improve the manuscript draft.

1. Please provide a comparative assessment of the strategies presented for the development of novel NNRTIs.

2. Please add a "take home" message summarising all the findings of the review. In particular, the Perspective section needs to be restructured to provide a more specific view of the authors on the state of the art strategies, their applications and perspectives.

3. I suggest you consider English language editing with the help of native speakers or translators with English language proficiency.

English language editing with the help of native speakers or translators with proficiency level of English is required.

Author Response

Reviewer 1

Dear Authors.

The manuscript contains a thorough review of the strategies used in drug discovery in general and provides appropriate examples of their application to the design of NNRTIs.

I think that in its current form the manuscript contains a lot of chemical information that needs to be recomposed to improve the summary for each of the strategies considered and a comparison of these strategies in the development of (a) drugs in general and (b) NNRTIs. In particular, it would be great if the authors could include the number of drugs, and especially NNRTIs, whose activity has been confirmed in experimental studies for each of the strategies and discuss the results. An additional comparative study of clinical trials for the development of NNRTIs would be an additional feature of the manuscript.

The volume can be reduced by focusing on the critical evaluation of the strategies and keeping the details that allow the reader to see the relationship between the strategy and its critical evaluation provided by the authors.

Thank you for the positive and critical comments for improvising the manuscript

Therefore, I suggest the following steps to improve the manuscript draft.

  1. Please provide a comparative assessment of the strategies presented for the development of novel NNRTIs.

Thanks. In the revised version of the manuscript we have included a new section for comparative assessment.

  1. Please add a "take home" message summarising all the findings of the review. In particular, the Perspective section needs to be restructured to provide a more specific view of the authors on the state of the art strategies, their applications and perspectives.

Thanks for the suggestion. We have updated the perspective section by summarizing the take home message.

  1. I suggest you consider English language editing with the help of native speakers or translators with English language proficiency.

English language editing with the help of native speakers or translators with proficiency level of English is required.

Thanks. The manuscript has revised completely for the English language correction and grammatical changes.

Reviewer 2 Report

This review focuses on the identification of emerging novel scaffolds of NNRTI and various approaches in the fields of medicinal chemistry, pharmacology, and formulation techniques for optimizing the lead molecule to a promising drug candidate. This comprehensive review has unique properties searching for drug candidates with different insights and could be helpful for researchers working at this area. This study can be accepted after minor revision.

Here are some points:

“Till date, thousands of people lose their lives annually from HIV infection, and continues to be a big public health issue globally” must be “Till date, thousands of people lose their lives annually from HIV infection, and HIV infection continues to be a big public health issue globally”.

Authors can add the chemical structures of Nevirapine (NVP), Delavirdine (DLV), Efavirenz (EFV), Etravirine 64 (ETR), Rilpivirine (RPV) and Doravirine (DOR).

-“ for optimizing the lead molecule to a promising drug candidate.” must be “for optimizing the lead molecule as a promising drug candidate”.

-“ Molecular hybridization is one of the important and successful strategies” must be “Molecular hybridization is one of the most important and successful strategies”.

-“ found no mortality” in Line 125 must be changed.

-“ The cytotoxicity of compound 4 was significantly reduced” must be “The cytotoxicity of compound 4 reduced significantly” in Line 129.

Minor editing of English language required.

Author Response

Reviewer 2

This review focuses on the emerging novel scaffolds of NNRTI and various approaches in the fields of medicinal chemistry, pharmacology, and formulation techniques for optimizing the lead molecule to a promising drug candidate. This comprehensive review has unique properties searching for drug candidates with different insights and could be helpful for researchers working at this area. This study can be accepted after minor revision.

Thanks for the reviewer for positive comments and suggestion for modification.

Here are some points:

“Till date, thousands of people lose their lives annually from HIV infection, and continues to be a big public health issue globally” must be “Till date, thousands of people lose their lives annually from HIV infection, and HIV infection continues to be a big public health issue globally”.

We have incorporated in the manuscript.

Authors can add the chemical structures of Nevirapine (NVP), Delavirdine (DLV), Efavirenz (EFV), Etravirine 64 (ETR), Rilpivirine (RPV) and Doravirine (DOR).

Included in the manuscript as Figure 15.

-“ for optimizing the lead molecule to a promising drug candidate.” must be “for optimizing the lead molecule as a promising drug candidate”.

-“ Molecular hybridization is one of the important and successful strategies” must be “Molecular hybridization is one of the most important and successful strategies”.

-“ found no mortality” in Line 125 must be changed.

-“ The cytotoxicity of compound 4 was significantly reduced” must be “The cytotoxicity of compound 4 reduced significantly” in Line 129.

All the above corrections are included in the revised manuscript.

Reviewer 3 Report

The review titled “Drug discovery strategies in the design and development of non-nucleoside reverse transcriptase inhibitors (NNRTIs)” by Vanangamudi et al summarizes the different drug design approaches for suitable lead candidates targeting HIV-1 and its mutatents. The authors did extensive literature search, collected relevant articles and wrote the entire review article. They had described the drug design approaches in a detailed manner with suitable examples. These approaches are very useful for the scientific community in designing lead candidates. These approaches could be adopted not only to HIV-1 but other diseases as well wherever in designing of small molecules required. The article can be suitable for Viruses journal and can be published.

Author Response

Reviewer 3

The review titled “Drug discovery strategies in the design and development of non-nucleoside reverse transcriptase inhibitors (NNRTIs)” by Vanangamudi et al summarizes the different drug design approaches for suitable lead candidates targeting HIV-1 and its mutants. The authors did extensive literature search, collected relevant articles and wrote the entire review article. They had described the drug design approaches in a detailed manner with suitable examples. These approaches are very useful for the scientific community in designing lead candidates. These approaches could be adopted not only to HIV-1 but other diseases as well wherever in designing of small molecules required. The article can be suitable for Viruses journal and can be published.

Thanks the reviewer for the very positive feedback.

Round 2

Reviewer 1 Report

Dear authors,

thank you for the corrections.

I still think that the current version of the manuscript contains the extensively thorough description of chemical details relevant to the development of non-nucleoside reverse transcriptase inhibitors. 
I suggest further reduction of the manuscript volume focusing only on the details of chemical and biochemical strategies that are important for development of antiviral compounds with the brief description of their structure-activity relationship. 

Also, I suggest, that the Title is changed, for instance, “Drug discovery” phrase can be removed from it.

English has been improved, but should be checked carefully again before final processing.

Author Response

We would like to thank you for taking the time to review our manuscript on the development of non-nucleoside reverse transcriptase inhibitors. Your valuable feedback and insights are greatly appreciated, and we have carefully considered your comments.

We understand your concern regarding the extensive content of the current manuscript, however reducing the content will potentially impact the understanding of the readers and in particular researchers other than the field of NNRTIs. The SAR information are largely focused on the scaffolds rather than the strategies implemented from lead to the desired novel candidate, highly improved potency, and physicochemical properties of the molecules. The manuscript also delves into the SAR when deemed essential, as seen in the context of bioisosteric replacement (Section 2.2), structure-based optimization (2.7), targeting highly hydrophobic channel (2.11) Targeting the hydrophilic solvent-exposed region (2.12) and dual-site targeting (2.13)

In addition to this, I have taken your suggestion and modified the title accordingly. The new title, “Strategies in the design and development of non-nucleoside reverse transcriptase inhibitors (NNRTIs)" aims to reflect the content of the manuscript while eliminating the term "Drug discovery" as recommended. Additionally, we revised completely for the English language correction and grammatical changes.

Thank you once again for your valuable feedback and your commitment to improve the manuscript's clarity and focus. We look forward to hear your thoughts on these revisions and any additional suggestions you may have.